# References Indeed Matter? Reference-Free Preference Optimization for Conversational Query Reformulation

## Abstract

Conversational query reformulation (CQR) has become indispensable for improving retrieval in dialogue-based applications. However, existing approaches typically rely on reference passages for optimization, which are *impractical* to acquire in real-world scenarios. To address this limitation, we introduce a novel *reference-free* preference optimization framework DUALREFORM that generates *pseudo* reference passages from *commonly-encountered* conversational datasets containing only queries and responses. DUALREFORM attains this goal through two key innovations: (1) *response-based inference*, where responses serve as proxies to infer pseudo reference passages, and (2) *response refinement via the dual-role of CQR*, where a CQR model refines responses based on the shared objectives between response refinement and CQR. Despite not relying on reference passages, DUALREFORM achieves 96.9–99.1% of the retrieval accuracy attainable only with reference passages and surpasses the state-of-the-art method by up to 31.6%.

## 1 Introduction

Retrieval-augmented generation (RAG) (Lewis et al., 2020; Asai et al., 2024; Zhang et al., 2024; Jeong et al., 2024) is frequently employed to integrate external knowledge into the generation process of large language models (LLMs). One of the main components is to retrieve the passage most relevant to a specific query from an external data source. For this purpose, *conversational query reformulation (CQR)* (Elgohary et al., 2019; Lin et al., 2020; Qian & Dou, 2022; Vakulenko et al., 2021; Wu et al., 2021; Ye et al., 2023) is often used to facilitate the retrieval of the most relevant passage by reformulating the raw query.

In CQR, a query is reformulated using a language model (LM) which has generally been trained on a target conversational dataset. The training dataset comprises a collection of queries and corresponding responses, with each query linked to the *reference passage* that represents the most ideal retrieval target (Anantha et al., 2021; Adlakha et al., 2022). As shown in Figure 1(a), preference optimization (Rafailov et al., 2024) leverages the preferences over the candidates for the best reformulated query, where the rank of the reference passage in the retrieved passages for each candidate dictates the candidate's preference. For instance, since the reference passage is ranked higher for the candidate Ⓐ than for other candidates, a CQR model is fine-tuned to produce queries akin to Ⓐ during inference.

However, this *reference-based* preference optimization (Yoon et al., 2024; Lai et al., 2025) relies on an *impractical* assumption that abundant reference passages are readily available. Most real-world conversational datasets, unfortunately, do not satisfy this assumption. Even worse, generating reference passages is very labor-intensive or expensive because annotators need to resolve coreferences (e.g., "its" in Figure 1(a)) and apply domain-specific knowledge (e.g., "gene editing").

Therefore, in this paper, we introduce a novel *reference-free* preference optimization framework, DUALREFORM, that eliminates the need for readily available reference passages. Instead, our approach generates *pseudo* reference passages from *commonly-encountered* conversational datasets, i.e., just a collection of queries and corresponding responses. Evidently, the primary challenge is accurately inferring pseudo reference passages, as the quality of pseudo supervision directly impacts model performance (Xie et al., 2020; Sohn et al., 2020; Amini et al., 2025). The novelty of our DUALREFORM framework lies in two ideas.

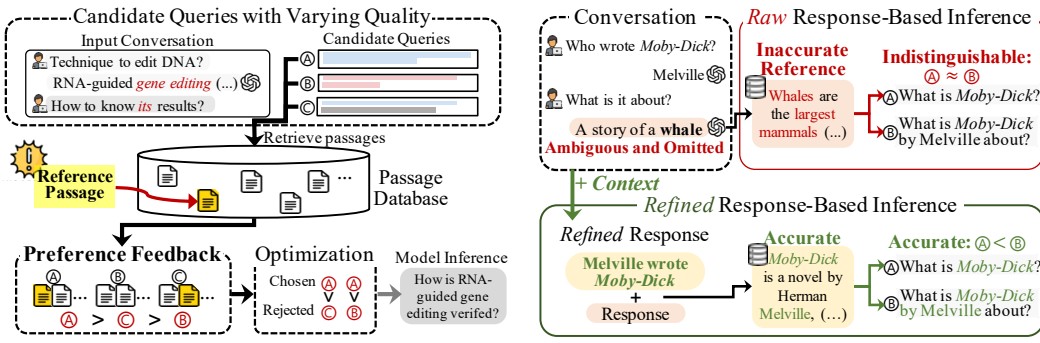

(a) Preference optimization framework for CQR.  (b) Key idea of DUALREFORM.

Figure 1: Overview of DUALREFORM. (a) Preference optimization framework for CQR with reference passages as a key component for generating preference feedback over candidate queries. (b) Key idea of DUALREFORM: Inferring pseudo reference passages through response refinement, addressing ambiguities and omissions in raw responses by incorporating conversational context.

(1) **Response-Based Inference**: While a (pseudo) reference passage is associated with a query, we propose to use its corresponding *response* to infer its pseudo reference passage. A pseudo reference passage for a query is an external piece of information that enhances the quality of its response. Note that, for the majority of conversational datasets employed in training CQR models, the responses are already accessible, despite not being optimized by these passages. So, why not utilize the responses to infer pseudo reference passages? We assert that the responses serve as excellent proxies or weak supervisions for pseudo reference passages.

(2) **Response Refinement by the *Dual-Role* of CQR**: Although the above idea appears intuitive, using raw responses may not yield accurate pseudo reference passages owing to potential ambiguities and omissions (e.g., "A story of whale" in Figure 1(b)). Thus, we propose to use *refined* responses rather than merely the raw responses. Such refined responses will clarify ambiguities and omissions by integrating pertinent conversational context (e.g., "Melvile wrote Moby-Dick" in Figure 1(b)).

Here, we exploit the *CQR model* to refine the raw responses. While the primary input for CQR is a query, this response refinement exactly aligns with the objective of CQR, namely, rephrasing a given query to enhance its relevance to the (pseudo) reference passage (Yoon et al., 2024). DUALREFORM leverages this *dual-role* of CQR, utilizing it for query reformulation during inference and for generating pseudo reference passages (through response reformulation) during training. One might argue that an LLM is suitable for this purpose, but we contend that taking advantage of the dual role offers numerous advantages. Most importantly, higher retrieval accuracy can be attained through preference optimization thanks to more accurate inference of pseudo reference passages. In addition, the overall procedure is simplified without an additional LLM; thus, monetary cost is saved by not relying on commercial LLMs.

In summary, DUALREFORM, featured by the dual-role of CQR, eliminates the need for reference passages, thereby enhancing its applicability to various conversational datasets. As far as we know, this is the *first* work that addresses reference-free preference optimization for CQR. Despite not using reference passages at all, DUALREFORM demonstrates a retrieval accuracy remarkably close (96.9–99.1%) to the optimal level only achievable with reference passages. Moreover, it outperforms the state-of-the-art methods with an average improvement of 15.7% in retrieval accuracy.

## 2 RELATED WORK

### 2.1 PREFERENCE OPTIMIZATION

Preference optimization methods (Schulman et al., 2017; Yang et al., 2024; Rafailov et al., 2024; Lou et al., 2024; Guo et al., 2024) aim to align the outputs of LMs with preferences by leveraging comparisons between outputs, such as rankings, rather than relying solely on labeled data or supervised targets. A prominent method is direct preference optimization (Rafailov et al., 2024), which optimizes LMs without relying on an explicit reward model, resulting in a computationally efficient and robust framework. Please refer to extensive surveys (Wirth et al., 2017; Jiang et al., 2024).

## 2.2 CONVERSATIONAL QUERY REFORMULATION

CQR methods typically employ LMs to generate self-contained queries by incorporating conversational contexts in three directions.

In prompt engineering methods, LLM-IQR (Ye et al., 2023) and LLM4CS (Mao et al., 2023) leverage LLMs to generate self-contained queries by carefully designing prompts that extract the relevant context from conversation. HyDE (Gao et al., 2023) extends this direction by generating synthetic passages related to the query.

In supervised fine-tuning methods, T5QR (Lin et al., 2020) fine-tunes a T5-base model (Raffel et al., 2020) for query reformulation using human-annotated reformulated queries. ConvGQR (Mo et al., 2023a) additionally fine-tunes an LM for query expansion, augmenting queries with potential responses, while aligning query embeddings to reference passages. However, they require creating high-quality reformulated queries, which are labor-intensive (Song et al., 2024) and often misalign with the retrieval objective (Yoon et al., 2024).

Addressing this issue, a preference optimization method such as RetPO (Yoon et al., 2024) aligns the LM with the retrieval objective by leveraging preferences among candidates for reformulated queries. AdaCQR (Lai et al., 2025) further enhances RetPo by incorporating multiple retrievers to strengthen preference signals. However, these methods assume the existence of abundant reference passages, prohibiting their utilization for most conversational datasets that lack such references. In Appendix A.1 and Table 9, we demonstrate that recently introduced large-scale conversational datasets (Zheng et al., 2023a; Ding et al., 2023; Köpf et al., 2023; Zhao et al., 2024) reflecting diverse real-world user interactions, which encompass approximately 7.3 million conversations, do *not* provide annotated reference passages.

## 3 PRELIMINARIES

### 3.1 CONVERSATIONAL QUERY REFORMULATION

A conversational session is represented as a sequence of query-response turns $\mathcal{T} = \{(x_t, a_t, G_t)\}_{t=1}^N$, where $x_t = (\mathcal{H}_{<t}, q_t)$ is the input comprising the query-response history $\mathcal{H}_{<t} = \{(q_i, a_i)\}_{i=1}^{t-1}$ and the current query $q_t$. Here, $a_t$ is the response to $q_t$, and $G_t = \{g_t^j\}_j$ is the set of reference passages to $q_t$.

Given an input $x_t$, CQR excutes a query reformulation function $\text{CQR}(\cdot; \theta)$, parametrized by a model $\theta$, to generate a self-contained query, which is then passed to a retrieval system $R(\cdot)$ to retrieve relevant passages, i.e., $\hat{G}_t = R(\text{CQR}(x_t; \theta))$. Formally, for a conversation session $\mathcal{T}$, the goal of CQR is to maximize the reformulation quality,

$$J_\theta(\mathcal{T}) = \frac{1}{|\mathcal{T}|} \sum_{t=1}^{|\mathcal{T}|} \mathcal{M}(\hat{G}_t, G_t), \tag{1}$$

where $\mathcal{M}(\hat{G}_t, G_t)$ is a metric (e.g., Recall@$k$) that evaluates the retrieval quality by comparing the retrieved passages $\hat{G}_t$ with $G_t$.

### 3.2 REFERENCE-BASED PREFERENCE OPTIMIZATION FOR CQR

**Preference Feedback Generation.** Preference optimization methods rely on the reference passages $G_t$ to produce *preference feedback*. For a given input $x_t$, an LLM is used to produce a set of candidate query reformulations $\{\tilde{q}_t^i\}_{i=1}^M$ with varying quality. The preference feedback $\text{pref}(G_t)$ is then defined as a sorted list of these candidates based on their relevance to the reference passages $G_t$,

$$\text{pref}(G_t) = \text{sort}\big(\{\tilde{q}_t^i\}_{i=1}^M, \text{ by decreasing } s(\tilde{q}_t^i \mid G_t)\big), \tag{2}$$

where $s(\tilde{q}_t^i \mid G_t)$ is the retrieval score, indicating how accurately $\tilde{q}_t^i$ retrieves passages containing $G_t$.

**Preference Optimization.** Preference optimization proceeds through two steps: supervised fine-tuning (SFT) and direct preference optimization (DPO). First, the SFT step trains the model $\theta$ on the top-ranked one $\tilde{q}_t^1$ from $\text{pref}(G_t)$, by minimizing the negative log-likelihood,

$$\ell_{\text{sft}}(x_t, G_t; \theta) = -\mathbb{E}_{\tilde{q}_t^i \sim \text{pref}(G_t)} \big[ \mathbb{1}_{[i=1]} \log P(\tilde{q}_t^i \mid x_t; \theta) \big], \tag{3}$$

where $P(\tilde{q} \mid x; \theta)$ is the probability of $\tilde{q}$ given the input $x$. Next, the DPO step optimizes the model to learn pairwise preferences from reformulation pairs $(\tilde{q}_t^i, \tilde{q}_t^j)$ such that $i < j$ in $\mathrm{pref}(G_t)$, by maximizing the preference likelihood,

$$\ell_{\mathrm{pref}}(x_t, G_t; \theta) = \mathbb{E}_{\tilde{q}_t^i, \tilde{q}_t^j \sim \mathrm{pref}(G_t)} \left[ \mathbb{1}_{[i<j]} \mathrm{r}(\tilde{q}_t^i, \tilde{q}_t^j; x_t, \theta) \right], \tag{4}$$

where $\mathrm{r}(\tilde{q}_t^i, \tilde{q}_t^j; x_t, \theta)$ represents the likelihood that the model $\theta$ ranks $\tilde{q}_t^i$ higher than $\tilde{q}_t^j$.

# 4 DUALREFORM: "REFERENCE-FREE" PREFERENCE OPTIMIZATION

## 4.1 PROBLEM STATEMENT

Our *reference-free* preference optimization framework, DUALREFORM, accommodates *commonly-encountered* scenarios where a conversation $\mathcal{T}_U = \{(x_t, a_t)\}_{t=1}^N$ does *not* include reference passages $G_t$. Instead, DUALREFORM generates *pseudo* reference passages $\tilde{G}_t$ to establish $\mathcal{T}_P = \{(x_t, a_t, \tilde{G}_t)\}_{t=1}^N$ to enable preference optimization. Then, the key challenge is how to accurately build the set of pseudo reference passages $\tilde{\mathcal{G}} = \{\tilde{G}_t\}_{t=1}^N$ such that the preference-optimized model $\theta_{\tilde{\mathcal{G}}}$ maximizes the retrieval performance on a target dataset, i.e.,

$$\tilde{\mathcal{G}}^* = \arg\max_{\tilde{\mathcal{G}}} J_{\theta_{\tilde{\mathcal{G}}}}(\mathcal{T}_P). \tag{5}$$

## 4.2 RESPONSE REFINEMENT BY CQR'S DUAL-ROLE

Because using raw responses harms the quality of pseudo reference passages, we leverage the CQR model not only for query reformulation but also for *response refinement*, thus introducing its *dual role*. As shown in Figure 2, the CQR model identifies and integrates the key context (e.g., "*Moby-Dick*") from the conversation (Mo et al., 2023a; Yoon et al., 2024), demonstrating a capacity advantageous for both

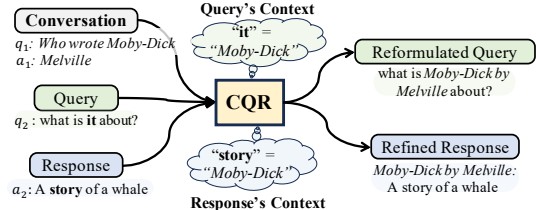

Figure 2: Dual-role of the CQR model.

query reformulation and response refinement. This dual role is a natural extension arising from the inherent alignment between the objective of response refinement and the retrieval objective in Eq. (1), where both aim to maximize the relevance to underlying reference passages.

We validate the effectiveness of the CQR's dual-role by comparing *single-* and *dual-role* configurations. As presented in Table 1, the single-role configuration employs an LLM for response refinement, whereas the dual-role configuration utilizes the CQR model.

**Theoretical Evidence.** Refining responses with a CQR model, optimized for high retrieval accuracy of reference passages, yields more accurate pseudo reference passages, thereby improving the CQR's retrieval accuracy. We provide a formal justification of this optimization using the generalization bound from the pseudo label denoising theory (Wei et al., 2021).

Table 1: Comparison of single- and dual-role configurations. LLM and CQR use a common language model.

| Roles | Single Role
Llama | Dual Role
DUALREFORM |
|---|---|---|
| **Response Ref.** | LLM | CQR |
| Query Ref. | | CQR |

**Assumption 4.1** (EXPANSION AND SEPARATION (WEI ET AL., 2021)). *We assume (i) c-expansion: each example is, on average, reachable to $c$ neighbors, i.e., $\mathbb{E}_x[|\mathcal{N}(x)|] = c$ with $c > 3$, where $\mathcal{N}(x)$ denotes the neighborhood of $x$; and (ii) $\mu$-separation: the average proportion of neighbors requiring different reference passages is $\mu$, which is negligibly small (e.g., $1/poly(d)$, the inverse of a polynomial in dimension).*

These assumptions, commonly used in prior studies (Wei et al., 2021; Park et al., 2023; Cai et al., 2021a), consider data distributions in which the examples with the same reference passages are close by $c$-expansion whereas those with different reference passages are well separated by $\mu$-separation. Under these assumptions, we derive the training error bounds for the single- and dual-role CQR models in Lemmas 4.2 and 4.3.

**Lemma 4.2** (SINGLE-ROLE BOUND). *Suppose that Assumption 4.1 holds. Then, the training error of a CQR model $\theta_{single}$ trained under the single-role configuration is bounded by the quality of pseudo reference passages generated by the LLM such that*

$$Err(\theta_{single}) \leq \frac{2}{c-1}Err(\theta_{LLM}) + \frac{2c}{c-1}\mu. \tag{6}$$

**Lemma 4.3** (DUAL-ROLE BOUND). *Suppose that Assumption 4.1 holds. Then, the training error of a dual-role CQR model $\theta_{dual}$ under the dual-role configuration is bounded by*

$$Err(\theta_{dual}) \leq \left(\frac{2}{c-1}\right)^2 Err(\theta_{LLM}) + \left(\frac{2c}{c-1}\right)\left(\frac{c+1}{c-1}\right)\mu. \tag{7}$$

These bounds extend the pseudo label denoising theorem (Wei et al., 2021), with complete proofs presented in Appendix B.1. Combining these lemmas, we compare the two configurations in Theorem 4.4.

**Theorem 4.4** (ERROR BOUND DIFFERENCE). *Let $\overline{Err}(\theta)$ denote the theoretical upper bound on the training error for the model $\theta$. Under Assumption 4.1, the bound of the dual-role configuration is smaller than that of the single-role configuration, i.e., $\overline{Err}(\theta_{dual}) < \overline{Err}(\theta_{single})$.*

*Proof.* Because $\mu$ is negligible and $(\frac{2}{c-1})^2 < \frac{2}{c-1}$ for $c > 3$, the dual-role configuration achieves a smaller upper bound. The complete proof is available in Appendix B.2. ∎

**Empirical Evidence.** Figure 3 empirically supports Theorem 4.4, showing that the refined responses by the CQR model yield more accurate pseudo references and higher retrieval accuracy than those by the single-role variants. *Pseudo reference accuracy*, in Figure 3(a), assesses response refinement by measuring the agreement between pseudo and ground-truth[1] reference passages; and *retrieval accuracy* in Figure 3(b), as defined in Eq. (1), assesses query reformulation by comparing passages retrieved via CQR against ground-truth reference passages. In the next section (see Figure 5), we will further demonstrate that the CQR model focuses on the context relevant to the response, while the single-role variant introduces less relevant context (e.g., including already mentioned movies when asking for unmentioned ones).

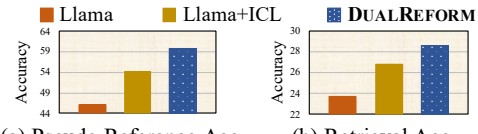

(a) Pseudo Reference Acc.    (b) Retrieval Acc.

Figure 3: Empirical comparison of single- and dual-role variants. All variants employ Llama3.1-8b-inst as their backbones and use the prompt detailed in Prompt 1. *Llama+ICL* variant additionally employs *in-context learning*.

### 4.3 OVERVIEW OF DUALREFORM

Figure 4 illustrates the reference-free preference optimization framework of DUALREFORM, driven by the CQR's dual role. It iteratively alternates between the two roles: as a *response refiner*, the CQR model helps generate pseudo reference passages $\tilde{G}_t$ (§ 4.4), which subsequently guide the optimization of the CQR model $\theta$ as a *query reformulator* to better align with the retrieval objective (§ 4.5). The improved CQR model is reintroduced for further response refinement, forming a self-reinforcing cycle (Amini et al., 2025) that continually enhances both pseudo references and retrieval performance. Algorithm 1 details each step.

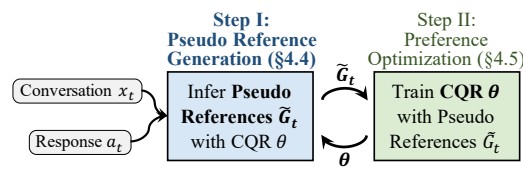

Figure 4: Overall flow of DUALREFORM.

### 4.4 STEP I: PSEUDO REFERENCE GENERATION

For the $t$-th conversation turn, we define *pseudo reference passages* $\tilde{G}_t$ in Definition 4.5.

**Definition 4.5** (PSEUDO REFERENCE PASSAGES). *Given a response $a_t$ at the $t$-th conversation turn, and its refined counterpart $\tilde{a}_t$, the pseudo reference passages $\tilde{G}_t$ are a set of passages, retrieved by $R(\cdot)$, which are most relevant to $a_t$ and $\tilde{a}_t$. Formally,*

$$\tilde{G}_t = R(a_t \,||\, \tilde{a}_t), \tag{8}$$

*where $||$ denotes string concatenation of $a_t$ and $\tilde{a}_t$.*

---

[1]The ground-truth information is used only for evaluation purposes, but it is *not* used in DUALREFORM.

**Response Refinement via CQR.** Since the CQR model expects a query format as its input, each raw response $a_t$ is converted into a query using a template function $T(\cdot)$. Formally, given an input $x_t = (\mathcal{H}_{<t}, q_t)$ and a raw response $a_t$, the refined response $\tilde{a}_t$ then becomes

$$\tilde{a}_t = \begin{cases} \text{CQR}\big((x_t, T(a_t)); \theta^*\big) & \text{if } \theta^* \text{ is trained} \\ \epsilon \text{ (i.e., empty string)} & \text{otherwise,} \end{cases} \quad (9)$$

where $\theta^*$ denotes the parameters of the trained CQR model from the previous iteration, held fixed in the current iteration. When $\theta^*$ is not sufficiently trained during the initialization period, an empty string $\epsilon$ is returned and no refinement is applied.

**Query-Forming Template Function.** When converting a response into a query format, we exploit the CQR model's mechanism as well. In addition to query reformulation, recent CQR models (Yoon et al., 2024; Mo et al., 2023a) also perform query expansion that adds a potential response. To harness the overall mechanism, we design the template function $T(\cdot)$ for a given response $a_t$ in Prompt 1.

This template prompts the CQR model to extract the context pertinent to $a_t$ by rephrasing *"last response ($\{a_t\}$)"* and to append a potential response to *"state the main points"* of $a_t$.

---

**Algorithm 1** DUALREFORM

---

1: **Input**: Conversational dataset $\mathcal{T}_U$, CQR model $\theta$, Number of iterations $n_{\text{iters}}$
2: $i \leftarrow 0$
3: **while** $i < n_{\text{iters}}$ **do**
4:     /* PSEUDO REFERENCE GENERATION § 4.4 */
5:     $\mathcal{T}_P \leftarrow \emptyset$
6:     **for** each $(x_t, a_t) \in \mathcal{T}_U$ **do**
7:         $\tilde{a}_t \leftarrow$ "" /* Initialize as an empty string */
8:         **if** $i > 0$ **then**
9:             $\tilde{a}_t \leftarrow$ RefineResponse($x_t, a_t, \theta$) /* Eq. (9) */
10:         $\tilde{G}_t \leftarrow$ RetrieveReference($a_t, \tilde{a}_t$) /* Eq. (8) */
11:         $\mathcal{T}_P \leftarrow \mathcal{T}_P \cup \{(x_t, a_t, \tilde{G}_t)\}$
12:     /* PREFERENCE OPTIMIZATION § 4.5 */
13:     $\theta \leftarrow$ Optimize($\mathcal{T}_P, \theta$) /* Eq. (10) */
14:     $i \leftarrow i + 1$
15: **return** $\theta$

---

> **Prompt 1. Template $T(a_t)$**
>
> Can you clearly *state the main points* of the *last response* ($\{a_t\}$), contextualizing them and resolving coreferences?

## 4.5 STEP II: PREFERENCE OPTIMIZATION

Once pseudo reference passages $\{\tilde{G}_t\}_{t=1}^N$ are ready, DUALREFORM conducts preference optimization on $\mathcal{T}_P = \{(x_t, a_t, \tilde{G}_t)\}_{t=1}^N$, pairing each conversational turn with its corresponding $\tilde{G}_t$.

**Preference Feedback Generation.** We use pseudo reference passages $\tilde{G}_t$ to derive preference feedback $\text{pref}(\tilde{G}_t)$ by ranking the candidate query reformulations $\{\tilde{q}_t^i\}_{i=1}^M$ according to their retrieval scores as in Eq. (2). The retrieval score $s(\tilde{q}_t^i \mid \tilde{G}_t)$ quantifies how accurately $\tilde{q}_t^i$ retrieves passages comprising $\tilde{G}_t$. This score is derived from a combination of multiple retrieval metrics, including Recall@k, MRR, and NDCG, to account for distinct aspects of retrieval quality (Manning et al., 2008). See Appendix C for its complete definition.

**Preference Optimization.** We then apply the standard preference optimization process consisting of SFT and DPO, following Eq. (3) and Eq. (4), guided by the preference feedback. The training objective is to optimize $\theta_{\tilde{G}}$ such that

$$\theta_{\tilde{G}} = \arg\min_{\theta} \sum_{t=1}^{|\mathcal{T}_P|} \ell_{\text{pref}}(x_t, \tilde{G}_t; \theta) \quad (10)$$

where $\theta$ is initially set to the parameter values obtained by SFT, i.e., $\arg\min_{\theta} \sum_{t=1}^{|\mathcal{T}_P|} \ell_{\text{sft}}(x_t, \tilde{G}_t; \theta)$.

## 5 EVALUATION

### 5.1 EXPERIMENT SETTING

**Dataset Preparation.** We evaluate the efficacy of DUALREFORM in realistic CQR deployment scenarios where reference passages are *unavailable* for target conversational datasets. To reflect the diversity of target datasets in the real world, we conduct experiments on three benchmarks: *QReCC* (Anantha et al., 2021) and *TopiOCQA* (Adlakha et al., 2022), which focus on general-domain topics with similar conversational contexts, and *SciConvQA*, a new benchmark focusing on specialized scientific domains with diverse conversational contexts.

SciConvQA: This is our proprietary conversational dataset, constructed using scientific journal data[2] provided by the Korea Institute of Science and Technology Information[3], a government-funded research institute. The dataset follows the conversation generation protocol of TopiOCQA (Adlakha et al., 2022) and will be publicly released upon acceptance. Additional details, including an example conversation, and visual comparisons with QReCC and TopiOCQA, are provided in Appendix D.

**Algorithms.** We compare DUALREFORM against (i) *LLM-based* CQR methods, LLM-IQR (Ye et al., 2023), HyDE-LLM (Gao et al., 2023), and LLM4CS-CoT (Mao et al., 2023); (ii) *SFT-only* CQR methods, T5QR (Lin et al., 2020); and (iii) *reference-based* CQR methods, ConvGQR (Mo et al., 2023a), HyDE-FT (Gao et al., 2023), RetPo (Yoon et al., 2024) and AdaCQR (Lai et al., 2025). The first and second categories do not need reference passages, but the second category needs optimally reformulated queries, which are even more costly to obtain. HyDE is configured as either HyDE-LLM or HyDE-FT.

To run *reference-based* baselines on a target dataset *devoid of reference passages*, we pre-train their CQR models on a *source* dataset having reference passages and transfer the models to the target dataset. Two transfer scenarios are intended for a *large* domain gap between a source and a target, *QReCC (source) → SciConvQA (target)*, and a *small* domain gap, *QReCC (source) → TopiOCQA (target)*.

Additionally, we include the *Upper Bound (RetPo[†])* baseline that performs preference optimization by RetPo using genuine reference passages of each target dataset. This Upper Bound baseline is used to estimate the ideal performance of CQR for a target dataset, although it is not practically usable owing to the necessity of reference passages. Importantly, *none* of the baselines except Upper Bound accesses reference passages.

**Metrics.** We evaluate (1) *pseudo reference accuracy*, assessing the agreement between our pseudo and ground-truth reference passages; (2) *retrieval accuracy*, evaluating the agreement between CQR-retrieved passages and ground-truth reference passage; and (3) *response generation accuracy*, measuring the quality of LLM-generated responses with CQR-retrieved passages. We measure pseudo reference and retrieval accuracy using MRR, NDCG@3, and Recall@$k$ (Mo et al., 2023a; Yoon et al., 2024), and generation accuracy using LLMEval, ROUGE, and BertScore (Jeong et al., 2024; Rau et al., 2024; Zheng et al., 2023b). In particular, LLMEval provides a comprehensive assessment of both the similarity of generated responses to ground-truth references and their overall usefulness and correctness. More details on the evaluation metrics are provided in Appendix E.1.

**Retriever Systems.** Following (Mo et al., 2023a; Ye et al., 2023; Yoon et al., 2024), we employ BM25 (Robertson et al., 2009) for sparse retrieval and GTR (Ni et al., 2022) for dense retrieval. We additionally employ ANCE (Xiong et al., 2021), another widely used dense retriever, with its results reported in Appendix F.7.

**Implementation Details.** We train all baselines, except RetPo, using their official repositories. Due to the absence of released code, we implement RetPo by adopting our strategy for preference feedback generation and confirm that it achieves better performance than the original paper. For DUALREFORM, pseudo reference passages are updated once per epoch throughout three epochs, following (Xie et al., 2020). The top-3 relevant passages are chosen as pseudo reference passages for each conversation turn. The mean and standard error of three repetitions with different random initializations are reported. Additional implementation details are provided in Appendix E.2.

## 5.2 MAIN RESULTS

Table 2 compares DUALREFORM against CQR baselines for the two target datasets.

**Significance of Reference-Free Preference Optimization.** Both Upper Bound and RetPo employ preference optimization, yet they exhibit contrasting results depending on the availability of reference passages from target datasets. The Upper Bound achieves the strongest results by using reference passages, whereas RetPo struggles without them even when the target dataset shares similar conversational contexts (QReCC → TopiOCQA). Moreover, although AdaCQR enhances RetPo by combining sparse and dense retrievers to reinforce preference signals, it remains explicitly designed

---

[2]https://aida.kisti.re.kr/data/b22c73ed-fa19-47b0-87b3-a509df8380e5
[3]https://www.kisti.re.kr/

Table 2: Retrieval accuracy of DUALREFORM compared with representative CQR baselines. The best and second-best results (excluding Upper Bound) are highlighted in bold and underlined, respectively. *Upper Bound* indicates the accuracy achieved by applying preference optimization (RetPo) using the *genuine reference passages* from each target dataset, which represents the idealized performance.

| Data | Query Reformulations | Sparse Retriever | | | | Dense Retriever | | | |
|---|---|---|---|---|---|---|---|---|---|
| | | MRR | NDCG | R@5 | R@20 | MRR | NDCG | R@5 | R@20 |
| SciConvQA | Upper Bound (RetPo†) | 20.89±0.30 | 18.91±0.58 | 30.09±0.62 | 43.45±0.50 | 23.73±0.70 | 22.49±0.80 | 31.95±1.00 | 45.14±0.67 |
| | Original Query | 4.85±0.00 | 4.36±0.00 | 6.70±0.00 | 10.85±0.00 | 6.24±0.00 | 5.57±0.00 | 8.17±0.00 | 14.18±0.00 |
| | LLM-IQR | 14.23±0.09 | 13.15±0.10 | 19.83±0.08 | 29.55±0.04 | 16.23±0.12 | 15.09±0.15 | 22.49±0.15 | 32.75±0.12 |
| | HyDE-LLM | 13.78±0.54 | 12.56±0.32 | 20.68±1.06 | 33.47±1.36 | 16.70±0.69 | 15.38±0.66 | 23.28±0.70 | 35.36±1.68 |
| | LLM4CS-CoT | 16.53±0.18 | 15.29±0.27 | 22.49±0.30 | 33.55±0.38 | 18.25±0.22 | 16.75±0.18 | 24.48±0.22 | 36.30±0.02 |
| | T5QR | 11.45±0.72 | 10.64±0.73 | 15.88±1.02 | 23.49±1.56 | 14.51±0.74 | 13.44±0.77 | 19.42±1.12 | 29.65±1.44 |
| | ConvGQR *(QReCC ↓ SciConvQA)* | 13.55±0.23 | 12.51±0.27 | 18.68±0.39 | 28.71±0.30 | 14.67±0.10 | 13.72±0.16 | 20.03±0.08 | 29.90±0.58 |
| | HyDE-FT | 11.09±0.47 | 9.85±0.52 | 15.89±1.07 | 27.74±1.74 | 12.16±1.03 | 11.09±1.15 | 16.63±1.90 | 27.50±2.65 |
| | RetPo | 15.60±0.23 | 14.36±0.24 | 21.61±0.33 | 32.63±0.50 | 17.95±0.08 | 16.74±0.08 | 24.31±0.36 | 35.45±0.56 |
| | AdaCQR | 15.97±0.31 | 13.97±0.34 | 23.00±0.49 | 36.91±0.58 | 20.14±0.17 | 18.83±0.19 | 27.49±0.23 | 40.22±0.06 |
| | **DUALREFORM** | **20.06**±0.19 | **18.42**±0.27 | **28.64**±0.28 | **43.14**±0.20 | **23.53**±0.09 | **22.04**±0.23 | **31.19**±0.18 | **45.21**±0.22 |
| TopiOCQA | Upper Bound (RetPo†) | 29.19±0.30 | 27.19±0.22 | 39.72±0.87 | 56.89±0.97 | 40.58±0.22 | 39.15±0.25 | 53.42±1.04 | 72.69±1.22 |
| | Original Query | 2.09±0.00 | 1.77±0.00 | 2.90±0.00 | 5.21±0.00 | 5.95±0.00 | 5.52±0.00 | 8.35±0.00 | 12.05±0.00 |
| | LLM-IQR | 17.21±0.09 | 15.66±0.11 | 24.34±0.11 | 37.72±0.06 | 32.53±0.13 | 33.26±0.16 | 45.61±0.18 | 61.63±0.14 |
| | HyDE-LLM | 20.65±1.29 | 18.59±0.87 | 28.26±1.95 | 44.34±1.03 | 33.39±1.48 | 32.02±1.34 | 45.54±0.05 | 61.49±0.46 |
| | LLM4CS-CoT | 26.81±0.66 | 25.40±0.38 | 37.12±1.12 | 53.36±1.80 | 37.55±0.75 | 35.92±1.04 | 52.45±0.35 | 69.51±0.04 |
| | T5QR | 12.14±0.11 | 10.39±0.12 | 17.61±0.10 | 31.16±0.17 | 28.36±0.07 | 27.42±0.05 | 39.78±0.07 | 53.21±0.07 |
| | ConvGQR *(QReCC ↓ TopiOCQA)* | 12.86±0.16 | 11.28±0.12 | 17.89±0.23 | 30.91±0.47 | 22.58±0.14 | 21.33±0.19 | 33.60±0.07 | 48.20±0.26 |
| | HyDE-FT | 11.22±1.70 | 10.05±1.70 | 16.25±3.10 | 25.38±4.20 | 19.28±0.63 | 17.80±0.80 | 27.01±0.88 | 38.85±1.38 |
| | RetPo | 23.20±0.33 | 21.41±0.28 | 32.18±0.20 | 48.54±0.22 | 35.67±0.05 | 34.28±0.04 | 49.99±0.31 | 67.47±0.10 |
| | AdaCQR | 24.78±0.63 | 22.60±0.78 | 35.90±0.52 | 52.11±0.64 | 36.45±0.50 | 35.20±0.56 | 50.29±0.52 | 67.71±0.43 |
| | **DUALREFORM** | **28.39**±0.39 | **26.08**±0.46 | **39.57**±1.05 | **56.62**±1.24 | **40.14**±0.21 | **38.33**±0.28 | **53.78**±0.58 | **71.99**±0.30 |

to exploit reference passages; consequently, its performance without them is notably lower than that of the Upper Bound. These findings call for an effective approach to preference optimization in reference-free scenarios.

**DUALREFORM: An Effective Reference-Free Preference Optimization Framework.** DUALREFORM consistently outperforms the baselines and achieves performance close to the Upper Bound across datasets and retrieval systems. On average, it achieves improvement of 15.70% over LLM4CS-CoT, the strongest baseline, and reaches 98.23% of the Upper Bound's performance. This result indicates the efficacy of DUALREFORM as a reference-free preference optimization approach.

**Robustness across Diverse CQR Domains.** DUALREFORM maintains strong performance across general domains (TopiOCQA) and specialized domains (SciConvQA), while the performance of the baselines varies considerably per domain. For example, DUALREFORM outperforms LLM4CS-CoT by 5.12% on TopiOCQA, and larger improvement of 26.28% on SciConvQA. This result reveals the domain sensitivity of the baselines and highlight DUALREFORM's robust effectiveness for diverse CQR domains, facilitated by the reference-free preference optimization on target datasets. More results on QReCC are provided in Appendix F.1.

## 5.3 ANALYSIS OF PSEUDO REFERENCE GENERATION

### 5.3.1 EFFECT OF RESPONSE REFINEMENT THROUGH CQR'S DUAL ROLE

Table 3 builds upon the analysis in Figure 3, comparing DUALREFORM and its single-role variants. Overall, these variants underperform compared to DUALREFORM, with Llama+ICL exhibiting declines of 11.62% in pseudo reference accuracy and 5.07% in retrieval accuracy. This result indicates DUALREFORM's capability to accurately refine responses for pseudo reference generation by leveraging the alignment between response refinement and retrieval objective. More results are provided in Appendix F.2.

Table 3: Comparison of response refinement methods.

| Data | Refine Methods | Pseudo Acc. | | Retrieval Acc. | |
|---|---|---|---|---|---|
| | | MRR | R@5 | MRR | R@5 |
| SciConvQA | Llama | 36.55 | 46.13 | 17.39 | 23.69 |
| | Llama+ICL | 44.82 | 54.13 | 19.02 | 26.77 |
| | DUALREFORM | **50.05** | **59.75** | **20.06** | **28.64** |
| TopiOCQA | Llama | 35.95 | 42.38 | 25.80 | 37.19 |
| | Llama+ICL | 50.53 | 59.31 | 27.92 | 37.79 |
| | DUALREFORM | **56.50** | **66.79** | **28.39** | **39.57** |

| Method | Refined Responses $\tilde{a}_9$ | Conversation |
|---|---|---|
| **Reference Passage** | Marvel Studios is known for the production of the Marvel Cinematic Universe films, …. within the Marvel Cinematic Universe, from "Iron Man" (2008) to "Spider-Man: Far From Home" (2019). … In addition to the Marvel Cinematic Universe, … the X-Men and Spider-Man … | **< Conversation History >** $q_1$: who plays the general in the incredible hulk? ... |
| Llama+ICL | The last answer in this conversation lists several production-related details about the movie "The Incredible Hulk". It … mentions … Iron Man, X-Men, and Spider-Man. | $q_3$: what is the storyline? $a_3$: Bruce Banner … in a military scheme to reinvigorate the "Super-Soldier" program through gamma radiation. …; |
| No Response | Can you clearly state the main points of the last answer, contextualizing them and resolving coreferences? … Marvel Studios produced the movie The Incredible Hulk, which … had a storyline where Bruce Banner … in a military scheme to reinvigorate the "Super-Soldier" program through gamma radiation. …. Marvel Studios … Iron Man, X-Men, and Spider-Man. | ... $q_8$: which production company is associated with it? $a_8$: Marvel Studios. $q_9$: can you name any other movies they produced? |
| **DUALREFORM** | Can you clearly state the main points of the movies produced by Marvel Studios, contextualizing them and resolving coreferences? … Marvel Studios include Iron Man, X-Men, and Spider-Man, …. Iron Man is …, X-Men is …, and Spider-Man is … All three films have been successful and have contributed to the Marvel Cinematic Universe. | **< Target Response >** $a_9$: Iron Man, X-Men, Spider-Man. |

Figure 5: Examples of refined responses generated by different methods on TopiOCQA. Fragments strongly aligned with the reference passage are highlighted in blue, while fragments with weaker connections (e.g., off-topic elements referring to previous conversation topics) are marked in red.

### 5.3.2 EFFECT OF ITERATIVE OPTIMIZATION

Table 4 presents the effect of the iterative procedure in Figure 4. In general, iteratively alternating between pseudo reference generation and model optimization progressively improves pseudo reference accuracy and retrieval performance, with convergence observed at the third update. This result indicates the importance of the iterative procedure in exploiting the synergy between pseudo reference quality and model optimization.

### 5.3.3 EFFECT OF QUERY-FORMING TEMPLATE

Table 5 compares the effect of the query-forming template with its variants. *Variant 1* omits the query-forming process with Prompt 1, directly using raw responses in Eq. (9). *Variant 2* excludes the response "($\{a_t\}$)" in Prompt 1. *Variant 3* and *Variant 4* deactivate the effects of the phrases "state the main points" and "last response ($\{a_t\}$)", respectively, from the refined response generated by the complete version of Prompt 1.

Across all variants, performance consistently degrades compared to DUALREFORM. The decline in Variant 1 shows the importance of structuring responses into query forms to exploit CQR's effective reformulation. The result for Variant 2 highlights the significance of explicitly integrating the raw response into the template for response-relevant context extraction. Finally, the degradation in Variant 3 and Variant 4 indicates that the two phrases

Table 4: Effect of *iterative* optimization within DUALREFORM.

| Data | Pseudo Ref. Updates | Pseudo Acc. | | Retrieval Acc. | |
|---|---|---|---|---|---|
| | | MRR | R@5 | MRR | R@5 |
| SciConvQA | 1 | 38.28 | 47.73 | 17.55 | 26.05 |
| | 2 | 46.21 | 54.96 | 19.53 | 26.92 |
| | 3 | **50.05** | **59.75** | **20.06** | **28.64** |
| TopiOCQA | 1 | 39.33 | 44.14 | 25.87 | 37.43 |
| | 2 | 55.24 | 65.32 | 28.08 | 39.25 |
| | 3 | **56.50** | **66.79** | **28.39** | **39.57** |

Table 5: Effect of the query-transforming template on pseudo reference accuracy.

| Variants | SciConvQA | | TopiOCQA | | Degrade |
|---|---|---|---|---|---|
| | MRR | R@5 | MRR | R@5 | |
| 1. w/o Prompt 1 | 45.38 | 54.35 | 54.57 | 64.16 | *6.97%* |
| 2. w/o "($\{a_t\}$)" in Prompt 1 | 47.29 | 57.35 | 50.00 | 60.26 | *8.46%* |
| 3. w/o "state the main points" | 43.62 | 52.10 | 54.29 | 63.42 | *9.70%* |
| 4. w/o "last response $\{a_t\}$" | 46.64 | 55.69 | 47.16 | 56.41 | *13.20%* |
| **DUALREFORM** | **50.05** | **59.75** | **56.50** | **66.79** | - |

contribute complementary information crucial for effective refinement. Additional results are presented in Appendix F.4.

### 5.3.4 QUALITATIVE ANALYSIS

Figure 5 presents refined responses generated by DUALREFORM and its variants for a conversation from TopiOCQA. Compared to other variants, DUALREFORM demonstrates superior contextual understanding by extracting relevant context (e.g., "Marvel Studios") and adding details regarding the response (e.g., "Marvel Cinematic Universe"). In contrast, Llama+ICL and No Response often rely on the less relevant context (e.g., "Hulk"). Additional results are provided in Appendix F.5.

### 5.3.5 COMPUTATIONAL COST AND LATENCY ANALYSIS

DUALREFORM incurs additional computation only in the offline training pipeline, in exchange for removing the need for costly manual annotation of reference passages. Once the CQR model is trained, inference-time latency is identical to that of the underlying CQR baseline, since DUALREFORM does not invoke any extra LLM calls or additional retrieval steps at test time.

Table 6: Wall-clock time for the major components of DUALREFORM, measured on a single RTX A6000 GPU in hours (h), minutes (min), and seconds (s). The "A100 (est.)" columns approximate timings on an NVIDIA A100 GPU using a 2.02× speedup factor, following Massed Compute (2024).

| Main Components | (i) Response Refinement via CQR | | (ii) Retrieval of Pseudo References | | (iii) DPO Training | |
|---|---|---|---|---|---|---|
| | A6000 | A100 (est.) | Sparse (BM25) | Dense (GTR) | A6000 | A100 (est.) |
| Elapsed Time | 3.58 h (1.289 s/query) | 1.77 h (0.639 s/query) | 1.15 min | 2.42 min | 2.79 h | 1.43 h |

Compared to reference-based methods such as RetPO (Yoon et al., 2024), the extra cost in the training stage stems from three components: (i) response refinement via CQR (Eq. 9), (ii) retrievals to obtain pseudo reference passages (Eq. 8), and (iii) DPO training on the resulting pseudo reference passages (Eq. 10). As summarized in Table 6, the overall overhead required by these steps is *moderate* and largely determined by the available GPU resources. In particular, using a single NVIDIA RTX A6000 necessitates only an additional 6.41 hours, with stages (i) and (iii) accounting for the majority of the computational overhead.

## 5.4 EXTENDED ANALYSES

### 5.4.1 RESPONSE GENERATION ACCURACY

Table 7 reports the generation accuracy using passages retrieved by different CQR baselines on SciConvQA, with Llama-3.1-8b-instruct as the generator and BM25 as the retriever. DUALREFORM achieves superior accuracy compared to baselines, demonstrating the consequence of its enhanced retrieval performance for downstream generation. Additional results on TopiOCQA are provided in Appendix F.6.

Table 7: Response generation accuracy with passages retrieved by different CQR methods.

| CQR Methods | Generation Accuracy | | | | | |
|---|---|---|---|---|---|---|
| | LLMEval | ROUGE-1 | ROUGE-L | BertScore | LLMEval-H | LLMEval-C |
| LLM-IQR | 27.76 | 20.07 | 17.64 | 86.01 | 46.82 | 58.19 |
| HyDE-LLM | 29.77 | 22.18 | 19.39 | 86.25 | 54.85 | 66.89 |
| LLM4CS-CoT | 26.42 | 20.23 | 17.58 | 85.95 | 59.20 | 67.22 |
| T5QR | 28.43 | 19.96 | 17.76 | 85.74 | 44.82 | 60.87 |
| ConvGQR | 23.08 | 20.81 | 18.37 | 85.95 | 51.51 | 62.88 |
| HyDE-FT | 23.17 | 19.81 | 17.40 | 86.02 | 49.81 | 66.80 |
| RetPo | 27.09 | 21.63 | 18.59 | 86.17 | 52.17 | 63.21 |
| **DUALREFORM** | **34.45** | **24.37** | **21.41** | **86.33** | **62.54** | **70.57** |

### 5.4.2 EXTENSION OF PREFERENCE OPTIMIZATION VIA SELF-GENERATED REFORMULATIONS

A persistent challenge in preference optimization frameworks, including RetPO and our DUAL-REFORM, is their dependence on a predetermined set of reformulated queries generated by an external LLM. Such dependence often incurs significant costs and limits the diversity of potential reformulations. To investigate whether CQR models can instead leverage their own reformulations, we introduce *DualReform+1R-Self*, a variant in which the additional CQR training stage is driven by queries produced by the *trained CQR model itself*. For comparison, we also implement *DualReform+1R-GPT*, which allocates the *same* additional training budget but uses the existing reformulations from ChatGPT. As shown in Table 8, DualReform+1R-Self consistently outperforms DualReform+1R-GPT, achieving relative gains of up to 4.16%. These results indicate that training CQR models can be effectively guided by reformulations generated by the model itself.

Table 8: Retrieval accuracy of DUALREFORM with an additional preference optimization using ChatGPT-generated (*+1R-GPT*) and self-generated (*+1R-Self*) reformulations.

| CQR method | Sparse Retriever | | | | Dense Retriever | | | |
|---|---|---|---|---|---|---|---|---|
| | MRR | nDCG | R@5 | R@20 | MRR | nDCG | R@5 | R@20 |
| DUALREFORM | 28.39±0.39 | 26.08±0.46 | 39.57±1.05 | 56.62±1.24 | 40.14±0.21 | 38.33±0.28 | 53.78±0.58 | 71.99±0.30 |
| DUALREFORM+*1R-GPT* | 28.14±0.47 | 26.14±0.51 | 38.36±0.44 | 55.95±0.31 | 40.72±0.17 | 39.38±0.18 | 54.15±0.45 | 72.98±0.83 |
| DUALREFORM+*1R-Self* | **29.22**±0.17 | **26.93**±0.33 | 39.25±1.02 | **58.28**±0.41 | **42.27**±0.57 | **40.79**±0.65 | **56.21**±0.46 | **74.53**±0.17 |

## 6 CONCLUSION

We propose DUALREFORM, a novel reference-free preference optimization framework for CQR, which eliminates the reliance on reference passages. Fully taking advantage of the *dual-role* of CQR, DUALREFORM generates accurate *pseudo* reference passages to guide preference optimization. Empirical results demonstrate the broad applicability of DUALREFORM across diverse conversational domains, without reliance on reference passages; notably, it achieves performance near (e.g., 98.23%) the levels attainable only with reference passages. Overall, we believe that our work sheds light on the importance of practical CQR approaches for diverse real-world conversational scenarios.

## LIMITATIONS

One limitation of the proposed DUALREFORM framework is that it relies on a single-dimensional preference structure, which constrains its ability to account for multi-dimensional feedback such as conciseness. Specifically, in the context of CQR, retrieval effectiveness may not be the only factor shaping preferences. While DPO is widely used for preference optimization, this single-dimensional focus limits its effectiveness when multiple factors matter. Recent work, such as CPO (Guo et al., 2024) and Sequential Alignment (Lou et al., 2024), offers promising alternatives that may better address this limitation. Future work will investigate their potential with CQR preference learning.

## REPRODUCIBILITY STATEMENT

Section 5 and Appendix E.2 present the complete implementation details, including query generation, retrieval modules, response synthesis, and training hyperparameters, for both DUAL-REFORM and the baseline models. The source code repository is publicly available at `https://anonymous.4open.science/r/DualReform`. In addition, Section 4.2 and Appendix B provide all theoretical results with their assumptions and proofs. Finally, Appendix D details the SciConvQA data generation pipeline—including acquisition, cleaning, and formatting scripts—so that readers can replicate the experimental setup.

## ETHICS STATEMENT

This work primarily aims at generating pseudo reference passages directly from a question answering dataset itself, without relying on human annotators, posing no ethical concerns during training. In creating the SciConvQA benchmark, we adhere to a common LLM-based conversation generation protocol detailed in the prior literature (Adlakha et al., 2022). Therefore, we do not anticipate any ethical violations or neagtive societal consequences resulting from this work.

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

REFERENCES INDEED MATTER?
REFERENCE-FREE PREFERENCE OPTIMIZATION FOR
CONVERSATIONAL QUERY REFORMULATION

# A  DISCUSSIONS

## A.1  REFERENCE PASSAGE FOR CQR TRAINING: DEFINITIONS, LIMITATIONS, AND OUR REFERENCE-FREE SETTING

**Reference Passages as Core Supervision Signal.** In this work, *reference passages* (often termed "gold passages") are the primary supervision signal. Specifically, they are the target documents that define what a conversational search system *should* retrieve for each query. Recent CQR methods exploit these gold passages in various ways for training. For example, CHIQ-FT and related approaches (Mo et al., 2023b; 2024a;b) construct *relevance judgments* over historical turns by checking whether adding a turn improves retrieval of the gold passage for the current query, and preference–optimization methods such as RetPO (Yoon et al., 2024) derive preferences between candidate rewrites from their ability to retrieve the same gold passage.

More concretely, let $q_t$ be the current query and $G_t^*$ its reference passage. For each historical turn $(q_i, G_i^*)_{i<t}$, a relevance judgment is defined by whether including this turn improves retrieval of $G_t^*$. Let $\mathcal{M}(q, G)$ denote a retrieval score (e.g., Recall@$k$) between query $q$ and reference passage $G$ as in Eq. 1. Then the contribution of $(q_i, G_i^*)$ is given by

$$\Delta_t = \mathcal{M}\big((q_i, G_i^*, q_t),\, G_t^*\big) - \mathcal{M}\big(q_t, G_t^*\big), \tag{11}$$

and the turn is judged relevant if $\Delta_i > 0$. Computing $\Delta_i$ explicitly requires the reference passage $G_t^*$ as the retrieval target. Thus, reference passages come first as the ground truth for retrieval, and relevance judgments (or preferences) are secondary supervisory signals *derived from* them.

**Limitations of Reference Passage-Based Supervision.** Conversational search benchmarks such as QReCC and TopiOCQA provide reference passages, making them well-suited for the above reference-based supervision. However, they cover only about 17K conversations and are largely restricted to Wikipedia-style topics. In contrast, recent large-scale conversational datasets (e.g., WildChat (Zhao et al., 2024), UltraChat (Ding et al., 2023), LMSYS-Chat (Zheng et al., 2023a), ShareGPT (RyokoAI, 2023), ASST1 Köpf et al. (2023)) contain millions of in-the-wild conversations that better reflect real user behavior but do *not* annotate reference passages and therefore ***cannot directly support such reference-based training***. As summarized in Table 9, this creates a pronounced mismatch between small, carefully curated benchmarks with gold references and large, realistic conversational logs without them. Relying solely on the former risks poor generalization to diverse domains, intents, and conversational styles, while manually annotating gold passages at the scale of the latter would be prohibitively expensive.

Table 9: Recent large-scale conversational datasets versus conversational search benchmarks. While the latter provide gold reference passages (✓), they are much smaller in scale and mostly restricted to Wikipedia topics, whereas the former lack such annotations (✗) but better reflect real user interactions.

| Dataset | # Convs (K) | *Reference passages* | Characteristics |
|---|---|---|---|
| *Recent large-scale conversational datasets* | | | |
| WildChat-4.8M-Full | 4800 | ✗ | In-the-wild ChatGPT logs across diverse domains |
| UltraChat | 1400 | ✗ | Multi-turn dialogues mimicking user scenarios |
| LMSYS-Chat-1M | 1000 | ✗ | Chatbot Arena logs over multiple LLMs |
| ShareGPT | 90 | ✗ | User-shared ChatGPT conversations |
| OASST1 | 66 | ✗ | Crowd-sourced dialogues with human ratings |
| *Main conversational search benchmarks* | | | |
| QReCC | 13.7 | ✓ | Combined from QuAC, TREC CAsT, and NQ |
| TopiOCQA | 3.5 | ✓ | Topic-centric conversations grounded in Wikipedia |

**Our Reference-Free Setting and Relation to Prior CQR Methods.** Our method is designed precisely for this *reference-free* regime. Instead of assuming that gold passages are available, we infer *pseudo* reference passages directly from conversational data and then construct relevance judgments (or other supervisory signals) from them, enabling reference-based supervision on large, realistic conversations. This makes our approach complementary to CHIQ-FT and RetPO: those methods assume access to reference passages and focus on how to best exploit them, whereas we provide a mechanism to approximate the missing reference signal when only raw conversational data are available, extending the applicability of reference-based CQR training beyond small, carefully annotated benchmarks.

### A.2 CLARIFICATION OF THE TERM "REFERENCE-FREE"

At first glance, one might reasonably regard our method as still *reference-based*, since DUALREFORM is ultimately trained using pseudo reference passages rather than operating without any reference signal at all. In this work, however, the term *reference-free* is used to emphasize that DUALREFORM does *not* rely on *human-annotated reference passages*. DUALREFORM assumes a conversational setting in which no gold references are available and instead automatically induces pseudo reference passages using the given retriever and CQR model. This is closely analogous to *self-supervised or pseudo-labeling methods*, where models are trained on label-like targets that are produced automatically from the data and a model rather than supplied as external supervision; in that literature, such approaches are often described as "label-free" optimization (Ferreira et al., 2025). In this sense, our setting is *reference-free* where we remove the need for human-labeled reference passages while still exploiting automatically constructed pseudo targets for training.

## B PROOFS OF THEORETICAL RESULTS

### B.1 PROOFS OF LEMMAS 4.2 AND 4.3

The formal statement of Assumption 4.1 is given in Assumptions B.1 and B.2. Let $\mathcal{B}(x)$ denote the set of all possible augmentations of an input $x$ produced by a data augmentation function $\mathcal{A}(\cdot)$.

**Assumption B.1** (EXPANSION). *An example can reach $c$ neighboring instances on average. Formally,* $\mathbb{E}_x[|\mathcal{N}(x)|] = c$ *with* $c > 3$*, where* $\mathcal{N}(x) = \{x' : \mathcal{B}(x) \cap \mathcal{B}(x') \neq \emptyset\}$*.*

**Assumption B.2** (SEPARATION). *The average proportion of neighboring instances belonging to a different reference passage is $\mu$, which is negligibly small. Formally,* $\mathbb{E}_x[\mathbb{1}_{[\exists x' \in \mathcal{B}(x) \text{ such that } G^*(x) \neq G^*(x')]}] = \mu$*, where $G^*(x)$ indicates the ground-truth reference passage for $x$.*

**The Proof of Lemma 4.2.** Under these assumptions, the pseudo labeling theory (Wei et al., 2021) provides pseudo label correction guarantees, improving training accuracy by rectifying erroneous pseudo labels, as presented in Lemma B.3.

**Lemma B.3** (PSEUDO LABEL DENOISING BOUND (WEI ET AL., 2021)). *Suppose that Assumptions B.1 and B.2 hold. Then, the training error of any minimizer $\hat{\theta}$ under the pseudo labeler $\theta_{PL}$ is bounded by*

$$Err(\hat{\theta}) \leq \frac{2}{c-1} Err(\theta_{PL}) + \frac{2c}{c-1}\mu. \tag{12}$$

The full proof of this result is presented in Appendix A.1 of (Wei et al., 2021). In our setting, the minimizer corresponds to the CQR model under a single-role configuration, which employs an LLM as the pseudo labeler. Assigning $\hat{\theta} = \theta_{single}$ and $\theta_{PL} = \theta_{LLM}$ in Eq. (12) gives $Err(\theta_{single}) \leq \frac{2}{c-1} Err(\theta_{LLM}) + \frac{2c}{c-1}\mu$, which matches the single-role configuration bound in Lemma 4.2.

Additionally, following (Wei et al., 2021), the general preference optimization framework for the CQR model employs *input consistency regularization* in building preference pairs, ensuring consistent preferences across input transformations. Aligned with standard data augmentation practices in language-based tasks (Jiao et al., 2025; Yoon et al., 2024), this framework employs LLM-based augmentation to produce various paraphrases of query reformulations by prompting an LLM multiple times, as shown in Eq. (2). To enforce a substantial gap between winning and losing examples

in preference pairs, high-effectiveness reformulations (winners) and low-effectiveness reformulations (losers) are sampled, following (Song et al., 2023; Yuan et al., 2024; Xu et al., 2023). Each preference pair thus includes differently rephrased queries while preserving consistent preference ordering over these rephrased ones. Consequently, through preference optimization in Eq. (4), the CQR model learns to maintain consistent preference rankings across these input transformations, thereby demonstrating the principle of input consistency regularization.

**Proof of Lemma 4.3.** Recall that the dual-role configuration employs the CQR model, trained under the single-role configuration, as its pseudo labeler, while the single-role configuration uses the backbone LLM model as its pseudo labeler. By leveraging this cascaded structure, we establish the error bound of the dual-role configuration in terms of the backbone LLM's error bound.

First, the error bound for the dual-role configuration relative to the single-role CQR model is given by

$$Err(\theta_{dual}) \leq \frac{2}{c-1} Err(\theta_{single}) + \frac{2c}{c-1}\mu. \tag{13}$$

Next, applying the error bound of the single-role configuration in Eq. (6) to $Err(\theta_{single})$ in the right-hand side of Eq. (13), we obtain

$$\begin{aligned} Err(\theta_{dual}) &\leq \frac{2}{c-1}\big[\frac{2}{c-1} Err(\theta_{LLM}) + \frac{2c}{c-1}\mu\big] + \frac{2c}{c-1}\mu \\ &\leq \left(\frac{2}{c-1}\right)^2 Err(\theta_{LLM}) + \left(\frac{2c}{c-1}\right)\left(\frac{c+1}{c-1}\right)\mu, \end{aligned} \tag{14}$$

which completes the proof of Lemma 4.3.

**Empirical Evaluation of Assumptions.** The expansion and separation assumptions in Assumptions B.1 and B.2 (Wei et al., 2021) are defined at the level of the *population* distribution: the constants $c$ and $\mu$ are quantified over all measurable neighborhoods under the full population distribution rather than with respect to a finite empirical sample. In practice, however, we only observe *finite* benchmarks drawn from this distribution, so these population-level quantities are, in general, *not* directly identifiable from data (Cai et al., 2021b; Lang et al., 2022; Park et al., 2023; Deng et al., 2024; Mitsuzumi et al., 2024).

Nevertheless, to make the theoretical analysis more concrete, we therefore estimate $c$ and $\mu$ on our datasets under a *practically motivated relaxation* of the original conditions. The resulting estimates are reported in Table 10. All datasets *satisfy* $c > 3$ with a relatively small $\mu$, suggesting that they lie in the regime described by Assumptions B.1 and B.2. We conjecture that the somewhat larger $\mu$ observed on TopiOCQA reflects its higher topical diversity and the resulting sparsity of the empirical sample distribution.

Table 10: Empirical estimates of the $c$-expansion and $\mu$-separation constants under our relaxed notion of neighbors on the three conversational benchmarks considered in this work.

| Dataset | $c$ | $\mu$ |
|---|---|---|
| SciConvQA | 3.00010 | 0.00060 |
| TopiOCQA | 3.00002 | 0.01932 |
| QReCC | 3.00002 | 0.01568 |

Our relaxation proceeds as follows. We first need to define "neighbors" at the sample level. In the population-level definition, neighbors are examples that (i) share the same reference passage and (ii) lie within a radius $r$. In finite conversational datasets, however, examples with the same reference passage are much rarer than in the underlying population. We therefore relax *"same"* to *"similar"* reference passages: in SciConvQA, passages from the same article (which is split into multiple passages) are treated as similar; in TopiOCQA and QReCC, passages within the same topic are treated as similar, using the dataset's topic labels for TopiOCQA and labels derived from clustering passage embeddings for QReCC. Meanwhile, the values of $c$ and $\mu$ vary with the choice of radius $r$, but $r$ is *not* fixed *a priori* in the original analysis of the expansion and separation assumptions (Wei et al., 2021). Thus, we select the smallest radius $r$ such that $c > 3$, and then verify that the resulting $\mu$

is sufficiently small. This procedure yields $r = 0.3278$ for SciConvQA, $r = 0.3414$ for TopiOCQA, and $r = 0.3419$ for QReCC.

*Implementation details.* Distances between examples are computed in an embedding space obtained by encoding each example (current query plus the immediately preceding turn) using the sentence-transformer GTR-T5-large (Ni et al., 2022) with a maximum token length of 384.

### B.2 PROOF OF THEOREM 4.4

Let $\overline{\mathrm{Err}}(\theta)$ denote the theoretical upper bound on the training error for a configuration $\theta$. Under Assumption 4.1, where $\mu \approx 0$, we obtain approximate upper bounds for the respective errors. From Lemma 4.2 (Single-Role Bound), Eq. (6) becomes

$$\overline{Err}(\theta_{single}) \approx \frac{2}{c-1} \, Err(\theta_{LLM}). \tag{15}$$

From Lemma 4.3 (Dual-Role Bound), Eq. (7) becomes

$$\overline{Err}(\theta_{dual}) \approx \left(\frac{2}{c-1}\right)^2 Err(\theta_{LLM}). \tag{16}$$

Here, "$\approx$" indicates approximate upper bounds for the respective error terms. Comparing the two approximate upper bounds, we obtain

$$\left(\frac{2}{c-1}\right)^2 \overline{Err}(\theta_{LLM}) < \frac{2}{c-1} \overline{Err}(\theta_{LLM}) \quad \text{if and only if} \quad c > 3. \tag{17}$$

Therefore, for $c > 3$, as stipulated in Assumption 4.1, the error bound under the dual-role configuration is strictly smaller than the error bound of the single-role configuration. This completes the proof of Theorem 4.4.

## C DEFINITION OF RETRIEVAL SCORE

We evaluate candidate query reformulations $\{\tilde{q}_t^i\}_{i=1}^M$ using pseudo reference passages $\tilde{G}_t$ based on their retrieval scores. The retrieval score $s(\tilde{q}_t^i \mid \tilde{G}_t)$ of a candidate indicates how accurately $\tilde{q}_t^i$ retrieves the passages to contain $\tilde{G}_t$. Specifically, our assessment focuses on three dimensions: (1) *coverage* (Definition C.1), reflecting how *comprehensively* reference passages are retrieved within specific cutoffs; (2) *immediacy* (Definition C.2), reflecting how *early* a reference passage appears in the ranking; and (3) *concordance* (Definition C.3), reflecting how well the ranking *aligns* with the ideal relevance ordering of reference passages. Finally, these three scores are combined in a single weighted value (Definition C.4).

**Definition C.1.** (COVERAGE SCORE) *The coverage score $s_{\mathrm{cov}}(\tilde{q}_t^i \mid \tilde{G}_t)$ is computed as*

$$s_{\mathrm{cov}}(\tilde{q}_t^i \mid \tilde{G}_t) = \frac{1}{|K|} \sum_{k \in K} \mathrm{Recall@k}\big(R(\tilde{q}_t^i), \tilde{G}_t\big), \tag{18}$$

*where $K$ is a predefined set of cutoff values.*

**Definition C.2.** (IMMEDIACY SCORE) *The immediacy score $s_{\mathrm{imm}}(\tilde{q}_t^i \mid \tilde{G}_t)$ is computed as*

$$s_{\mathrm{imm}}(\tilde{q}_t^i \mid \tilde{G}_t) = \mathrm{MRR}\big(R(\tilde{q}_t^i), \tilde{G}_t\big). \tag{19}$$

**Definition C.3.** (CONCORDANCE SCORE) *The concordance score $s_{\mathrm{con}}(\tilde{q}_t^i \mid \tilde{G}_t)$ is computed as,*

$$s_{\mathrm{con}}(\tilde{q}_t^i \mid \tilde{G}_t) = \mathrm{NDCG}\big(R(\tilde{q}_t^i), \tilde{G}_t\big). \tag{20}$$

**Definition C.4.** (RETRIEVAL SCORE) *For each candidate $\tilde{q}_t^i$, the retrieval score $s(\tilde{q}_t^i \mid \tilde{G}_t)$ is a weighted sum of the coverage, immediacy, and concordance scores, as*

$$s(\tilde{q}_t^i \mid \tilde{G}_t) = \omega_1 s_{\mathrm{cov}}(\tilde{q}_t^i \mid \tilde{G}_t) + \omega_2 s_{\mathrm{imm}}(\tilde{q}_t^i \mid \tilde{G}_t) + \omega_3 s_{\mathrm{con}}(\tilde{q}_t^i \mid \tilde{G}_t), \tag{21}$$

*where $\omega_1, \omega_2, \omega_3 \geq 0$ and $\omega_1 + \omega_2 + \omega_3 = 1$.*

Details on the retrieval evaluation metrics are provided in Appendix E.1.

---

**SciConvQA conversation**

---

Q: How do genomic tools enhance animal breeding programs?
A: Genomic tools, such as single nucleotide polymorphism (SNP), have led to a new method known as "genomic selection."
   This method utilizes dense SNP genotypes covering the entire genome to predict the breeding value.
Q: Which application is used in evaluating these programs?
A: ZPLAN+ is used to evaluate and optimize these programs.
Q: What parameters does it consider?
A: It considers genetic and economic parameters.
Q: Can you tell me more about the types of strategies it models?
A: It models four selection strategies: the current conventional program and three based on genomic information.
Q: What's unique about the final approach among them?
A: The final approach, GS3, is unique because it combines pedigree, genomic enhanced breeding values,
   performance, and progeny information.
Q: How were the male candidates evaluated in this scheme?
A: They were genotyped and their selection was based on performance tests and GEBV.
Q: Do we have information about the costs associated with these methods?
A: Yes, the cost of genotyping was assumed to be $120 per pig.
Q: How does this cost compare to more traditional testing methods?
A: Performance testing costs $55 per tested pig, which is less than genotyping.
Q: How many candidates undergo this evaluation when adventurers start the process?
A: Initially, 1,000 male candidates were considered in the genomic selection process.
Q: In the final phase of selection, how many of these males are retained?
A: In the final phase, 23 senior boars were retained.
Q: How does the conventional program handle progeny information differently than genomics?
A: The conventional program uses progeny records without considering genetic marker information.
Q: Can you summarize which traits measured the field performance?
A: Traits measured include average daily gain, back fat thickness, and feed conversion rate.
Q: Are these the same for testing the station?
A: No, station tests focus on meat quality traits like pH, meat color (L*), and intra-muscular fat.
Q: In this optimized workflow, what benefit is sought above all?
A: The primary goal is high genetic gains with low breeding costs.

---

Table 11: Example of a SciConvQA conversation. The conversation is generated from Lopez et al. (2016), published in a renowned journal and stored as 'JAKO201614137726690.json' in the scientific journal dataset described in Section D.2.

# D DATASET DETAILS

## D.1 GENERAL-DOMAIN: QReCC AND TOPIOCQA

The QReCC dataset (Anantha et al., 2021) contains 14K multi-turn conversations with a total of 80K question-answer pairs, aiming to retrieve reference passages from a large corpus of 54 million passages. Similarly, the TopiOCQA dataset (Adlakha et al., 2022) includes 3.9K conversations featuring topic shifts, comprising 51K question-answer pairs. Its passage collection is derived from Wikipedia and consists of approximately 20 million passages. For both datasets, small random subsets of the training data were used to construct the validation sets. While these datasets are well-suited for general-domain conversational contexts, they lack focus on domain-specific applications such as scientific question answering.

## D.2 SPECIALIZED-DOMAIN: SCICONVQA

Information-seeking conversations span a wide range of domains, from general topics to specialized areas like science, reflecting diverse user interests. To evaluate existing CQR methods and DUAL-REFORM, we introduce the SciConvQA dataset, composed of information-seeking conversations generated from renowned scientific journals.

The conversation generation process follows the protocol described in Appendix A of the Topi-OCQA (Adlakha et al., 2022) paper, which provides the methodology for creating conversational datasets. While TopiOCQA relies on crowd-sourced annotations, it incurs high costs or risks of diminished quality when applied to specialized scientific domains. Hence, we employ gpt-4o-2024-08-06 (OpenAI, 2023) for automated conversation generation, followed by post-hoc manual quality validation. Overall, the conversation generation process involves two steps: (1) selecting a scientific

```
Create an information-seeking conversation between two annotators: a questioner and an answerer. We give you
multiple sections of an academic paper as seed topics for the information-seeking questions.
Assume that the questioner has access only to the main topic of the given content, while the answerer can
access the full text. Allow topic switching by enabling the answerer to refer to sections from different
sections of papers.

The annotators are provided with the following guidelines.
Guidelines for the questioner:
  • The first question should be unambiguous and clear about the main topic of academic paper. The
  questioner, only knowing the main topic, cannot ask directly about the study's focus. Thus, do not ask like
  "What is the primary focus of the study?" as 1st question.
  • The follow-up questions are contextualized and always dependent on the conversation history (especially,
  last answerer's respond) so that question itself is hard to understand.
  • Avoid using same words as in section titles of the document. E.g. if the section title is "Awards", a
  plausible question can be "What accolades did she receive for her work?".
  • The conversation should involve multiple documents (topics).

Guidelines for the answerer:
  • Based on the question, identify the relevant document and section.
  • The answer should be based on the contents of the identified document.
  • The rationale should be a sub-string of content such that it justifies the answer and should be recorded
  below the answers.
  • The answer should be a sub-string in rationale whenever possible. However, answers should be edited to
  fit the conversational context (adding yes, no), perform reasoning (e.g. counting) etc.
  • Personal opinions should never be included.

• Example:
  – Content: {content}
  – Information-seeking conversation: {conversation}

• An information-seeking conversation:
  – Content: {topic seed}

Write an information-seeking conversation comprising 10–15 QA turns based on the provided content. The
questions should employ co-references rather than explicit identifiers (e.g., names, titles, locations) to
ensure contextual dependence on prior turns. Do not replicate content from demonstrative examples.
```

Figure 6: Prompt used for SciConvQA dataset generation. The example of {content} is provided in Figure 7, and the example of {conversation} can be found in Figure 8.

journal as the seed topic and (2) generating questioner-answerer interactions. Table 11 shows a representative conversation from SciConvQA.

**Seed Topics and Document Collection.** SciConvQA is constructed using scientific journal data provided by the Korea Institute of Science and Technology Information (KISTI), a government-funded research institute. The scientific journal dataset, accessible at `https://aida.kisti.re.kr/data/b22c73ed-fa19-47b0-87b3-a509df8380e5`, includes a total of 481,578 academic articles, comprising both Korean and English publications. Detailed information about the dataset construction is available at the linked source. For our study, we utilize 120,916 English articles to construct the external database corpus, from which a subset is sampled to generate conversations. These articles span 749 diverse scientific fields, including biology, medicine, and architecture.

**Conversation Generation.** We modify the conversation annotation protocol of TopiOCQA to design a prompt for gpt-4o-2024-08-06, including an in-context demonstration to illustrate the conversation generation process based on a seed topic. The prompt template with its demonstration is shown in Figures 6–8. During the conversation generation process, each article serves as a seed topic, and the reference passages for conversation turns are selected from the article. Specifically, for each conversation turn, a reference passage ("rationale" in the prompt) is selected as a substring of the article's content that justifies the answer, recorded directly below the corresponding answer, as the demostrastive conversation in Figure 8.

**Passage Database Construction.** The passage database is constructed using 120,916 English articles as the retrieval target. Specifically, we employ Langchain's `RecursiveCharacterTextSplitter` with a chunk size of 500 and a chunk overlap of 100 (LangChain, 2025), resulting in a database consisting of 1,909,524 passages.

The Byzantine Empire, also referred to as the Eastern Roman Empire, was the continuation of the Roman Empire centered in Constantinople during Late Antiquity and the Middle Ages. The eastern half of the Empire survived the conditions that caused the fall of the West in the 5th century AD, and continued to exist until the fall of Constantinople to the Ottoman Empire in 1453. During most of its existence, the empire remained the most powerful economic, cultural, and military force in the Mediterranean world. The term "Byzantine Empire" was only coined following the empire's demise; its citizens referred to the polity as the "Roman Empire" and to themselves as "Romans". Due to the imperial seat's move from Rome to Byzantium, the adoption of state Christianity, and the predominance of Greek instead of Latin, modern historians continue to make a distinction between the earlier Roman Empire and the later Byzantine Empire.

The empire was largely dismantled in 1204, following the Sack of Constantinople by Latin armies at the end of the Fourth Crusade; its former territories were then divided into competing Greek rump states and Latin realms. Despite the eventual recovery of Constantinople in 1261, the reconstituted empire would wield only regional power during its final two centuries of existence. Its remaining territories were progressively annexed by the Ottomans in perennial wars fought throughout the 14th and 15th centuries. The fall of Constantinople to the Ottomans in 1453 ultimately brought the empire to an end. Many refugees who had fled the city after its capture settled in Italy and throughout Europe, helping to ignite the Renaissance. The fall of Constantinople is sometimes used to mark the dividing line between the Middle Ages and the early modern period.

The situation became worse for Byzantium during the civil wars after Andronikos III died. A six-year-long civil war devastated the empire, allowing the Serbian ruler Stefan Dušan to overrun most of the empire's remaining territory and establish a Serbian Empire. In 1354, an earthquake at Gallipoli devastated the fort, allowing the Ottomans (who were hired as mercenaries during the civil war by John VI Kantakouzenos) to establish themselves in Europe. By the time the Byzantine civil wars had ended, the Ottomans had defeated the Serbians and subjugated them as vassals. Following the Battle of Kosovo, much of the Balkans became dominated by the Ottomans.

Constantinople by this stage was underpopulated and dilapidated. The population of the city had collapsed so severely that it was now little more than a cluster of villages separated by fields. On 2 April 1453, Sultan Mehmed's army of 80,000 men and large numbers of irregulars laid siege to the city. Despite a desperate last-ditch defence of the city by the massively outnumbered Christian forces (c. 7,000 men, 2,000 of whom were foreign), Constantinople finally fell to the Ottomans after a two-month siege on 29 May 1453. The final Byzantine emperor, Constantine XI Palaiologos, was last seen casting off his imperial regalia and throwing himself into hand-to-hand combat after the walls of the city were taken.

Mehmed continued his conquests in Anatolia with its reunification and in Southeast Europe as far west as Bosnia. At home, he made many political and social reforms. He encouraged the arts and sciences, and by the end of his reign, his rebuilding program had changed Constantinople into a thriving imperial capital. He is considered a hero in modern-day Turkey and parts of the wider Muslim world. Among other things, Istanbul's Fatih district, Fatih Sultan Mehmet Bridge and Fatih Mosque are named after him.

Anatolia (Turkish: Anadolu), also known as Asia Minor, is a large peninsula or a region in Turkey, constituting most of its contemporary territory. Geographically, the Anatolian region is bounded by the Mediterranean Sea to the south, the Aegean Sea to the west, the Turkish Straits to the north-west, and the Black Sea to the north. The eastern and southeastern boundary is either the southeastern and eastern borders of Turkey, or an imprecise line from the Black Sea to Gulf of Iskenderun. Topographically, the Sea of Marmara connects the Black Sea with the Aegean Sea through the Bosporus strait and the Dardanelles strait, and separates Anatolia from Thrace in the Balkan peninsula of Southeastern Europe.

The Akkadian Empire (/əˈkeɪdiən/) was the first known ancient empire of Mesopotamia, succeeding the long-lived civilization of Sumer. Centered on the city of Akkad (/ˈækæd/) and its surrounding region, the empire united Akkadian and Sumerian speakers under one rule and exercised significant influence across Mesopotamia, the Levant, and Anatolia, sending military expeditions as far south as Dilmun and Magan (modern United Arab Emirates, Saudi Arabia, Bahrain, Qatar and Oman) in the Arabian Peninsula.

Figure 7: Example of {content} in Figure 6.

**Post-Processing.** To ensure compatibility with existing datasets (e.g., TopiOCQA), we standardize the format of the raw conversations generated during the conversation generation process. Due to inconsistencies in the output structure of chat completions, we extract only the relevant content using a custom post-processing pipeline. Furthermore, the passage ID for the ideal reference passage corresponding to each query is assigned by identifying the longest common substring between the generated "rationale" and passages in the external passage database. The entire implementation of the post-processing procedure is provided in the DUALREFORM's code repository.

### D.3 EXPLORATORY ANALYSIS FOR SCICONVQA

**Data Statistics.** Table 12 presents the statistics of the SciConvQA dataset. In summary, there are 11,953 turns across 900 conversations, with an average of 11.53 words per query, 15.59 words per response, and 13.28 turns per conversation.

```
Q1: when was the byzantine empire born what was it originally called?
A1: 5th century AD and was called Eastern Roman Empire, or Byzantium
rationale: The Byzantine Empire, also referred to as the Eastern Roman Empire, was the continuation of the
Roman Empire centered in Constantinople during Late Antiquity and the Middle Ages. The eastern half of the
Empire survived the conditions that caused the fall of the West in the 5th century AD, and continued to exist
until the fall of Constantinople to the Ottoman Empire in 1453.

Q2: and when did it fall?
A2: 1453
rationale: The fall of Constantinople to the Ottomans in 1453 ultimately brought the empire to an end.

Q3: which battle or event marked the fall of this empire?
A3: A six-year-long civil war followed by attack from Sultan Mehmed's army
rationale: A six-year-long civil war devastated the empire;  On 2 April 1453, Sultan Mehmed's army of 80,000
men and large numbers of irregulars laid siege to the city.

Q4: did he conquer other territories as well?
A4: Yes. Anatolia and in Southeast Europe as far west as Bosnia
rationale: Mehmed continued his conquests in Anatolia with its reunification and in Southeast Europe as far
west as Bosnia

Q5: where is the first area located in present day terms?
A5: Turkey
rationale: Anatolia (Turkish: Anadolu), also known as Asia Minor,[a] is a large peninsula or a region in
Turkey

Q6: who were the oldest known inhabitants of this region?
A6: Mesopotamian-based Akkadian Empire
rationale: The Akkadian Empire (/əˈkeɪdiən/)[2] was the first known ancient empire of Mesopotamia
```

Figure 8: Examples of {conversation} in Figure 6.

Table 12: Dataset statistics of SciConvQA

| Dataset | Train | Test | Overall |
|---|---|---|---|
| # Turns | 9,999 | 1,954 | 11,953 |
| # Conversations | 750 | 150 | 900 |
| # Words / Query | 11.51 | 11.62 | 11.53 |
| # Words / Response | 15.54 | 15.83 | 15.59 |
| # Turns / Conversation | 13.33 | 13.03 | 13.28 |

**Domain Similarity Comparison of Conversational Datasets.** The t-SNE visualization in Figure 9 offers a qualitative insight into the similarity among three conversational datasets: QReCC, TopiOCQA, and SciConvQA. In general, QReCC and TopiOCQA show significant overlap in the embedding space, suggesting high semantic similarity between these datasets. This is attributed to their common focus on general-domain conversational topics. In contrast, SciConvQA forms a clearly separate cluster due to its specialized domain (e.g., scientific topics), reflecting the domain difference from the other two datasets. In real-world applications, CQR models are required to handle such diverse datasets across general-domain and specialized-domain conversations.

# E    EXPERIMENT DETAILS

## E.1    EVALUATION METRICS

**Metrics for Retrieval Accuracy.** We employ three widely used metrics (Qu et al., 2020; Yu et al., 2021; Lin et al., 2021; Ye et al., 2023): MRR, NDCG@3, and Recall@$k$. MRR evaluates how well the system ranks the *first* relevant result, with higher scores indicating that a reference passage appears earlier in the ranked list. NDCG@3 measures the overall *alignment* of ranking with with the ideal relevance ordering of reference passages by prioritizing that they are positioned closer to the top of the retrieved passage list. Recall@$k$ assesses the *coverage*, capturing the fraction of reference passages retrieved within the top $k$ results. Together, these metrics provide a holistic assessment of the system's ability to perform accurate and relevant passage retrieval.

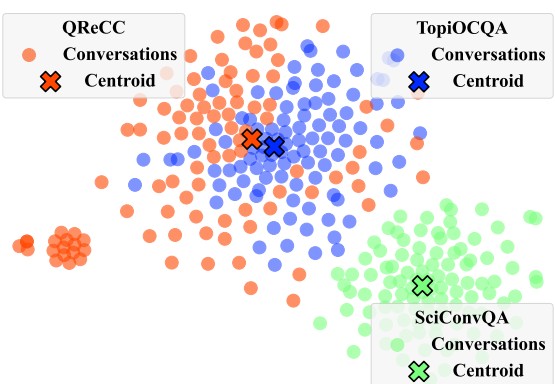

Figure 9: t-SNE visualization (Van der Maaten & Hinton, 2008) of three conversational datasets: QReCC, TopiOCQA, and SciConvQA. For each dataset, 100 randomly sampled conversations were encoded using the pretrained Sentence Transformer (Reimers & Gurevych, 2019; Ni et al., 2022; Ye et al., 2023) and projected into 2D embedding space via t-SNE. The conversations (denoted by ● symbols) from the same dataset are in the same color, where the centroid of each dataset's conversations is denoted by a ✕ symbol.

**Metrics for Generation Accuracy.** We employ widely used metrics (Zhang et al., 2024; Baek et al., 2023; Asai et al., 2024; Mallen et al., 2023; Jeong et al., 2024; Chirkova et al., 2024): LLMeval, ROUGE, and BertScore. LLMEval employs gpt-4o-2024-08-06 (OpenAI, 2024) as the evaluator, providing human-aligned relevance and generative quality assessments. ROUGE, specifically ROUGE-1 and ROUGE-L, evaluates lexical and structural alignment through unigram overlap and longest common subsequences. BERTScore (Zhang et al., 2020; Devlin et al., 2018) uses contextual embeddings to measure semantic similarity, enabling robust evaluation beyond surface-level matching.

### E.2 IMPLEMENTATION DETAILS

All baseline models are trained using their official repositories (Lin et al., 2020; Gao et al., 2023; Mo et al., 2023a; Ye et al., 2023; Mao et al., 2023), with the exception of RetPo, which we implement because no public code is available.

RetPo's hyperparameters are tuned via grid search for both SFT and DPO: SFT is trained for one epoch with a learning rate of $2 \times 10^{-5}$ and a batch size of 32, while DPO uses $\beta = 0.1$ (chosen from $\{0.1, 0.2, 0.3, 0.4, 0.5\}$) and trains for two epochs under the same learning rate and batch size. Our implementation of RetPo achieves improved performance over the authors' results.

We adopt the same SFT configuration for DUALREFORM but set $\beta = 0.5$ during DPO to mitigate overfitting to initial pseudo reference passages. For other hyperparameters, DUALREFORM updates pseudo reference passages every epoch, repeating this process three times following (Xie et al., 2020), and selects the top-3 relevant passages per query-response turn. Hyperparameter sensitivity analyis is provided in Appendix F.9.

For backbone models, LLM-IQR, HyDE-LLM, and LLM4CS-CoT use gpt-3.5-turbo-0125 (OpenAI, 2022), while RetPo, HyDE-FT, and DUALREFORM use Llama3.1-8b-instruct (Dubey et al., 2024). T5QR and ConvGQR follow their official T5-base implementations.

All models were implemented in PyTorch 2.1.2 and trained on NVIDIA RTX A6000 Ada GPUs. The source code is publicly available at https://anonymous.4open.science/r/DualReform.

**Details of Retrieval Systems.** We use Pyserini (Johnson et al., 2019) and Faiss (Johnson et al., 2019) for BM25 (Robertson et al., 2009) and GTR (Ni et al., 2022) retrieval systems, respectively. For BM25, we adopt the parameter settings from previous studies (Mo et al., 2023a; Yoon et al., 2024; Ye et al., 2023), configuring $k_1 = 0.82$, $b = 0.68$ for QReCC, and $k_1 = 0.9$, $b = 0.4$ for TopiOCQA and SciConvQA, where $k_1$ adjusts term frequency normalization and $b$ controls the impact of document

```
Given a question, its previous questions (Q) and answers (A), decontextualize the question by addressing
coreference and omission issues. The resulting question should retain its original meaning and be as
informative as possible, and should not duplicate any previously asked questions in the context.

Context: [Q: When was Born to Fly released? A: Sara Evans's third studio album, Born to Fly, was released on
October 10, 2000.]
Question: Was Born to Fly well received by critics?
Rewrite: Was Born to Fly well received by critics?

Context: [Q: When was Keith Carradine born? A: Keith Ian Carradine was born August 8, 1949. Q: Is he married?
A: Keith Carradine married Sandra Will on February 6, 1982.]
Question: Do they have any children?
Rewrite: Do Keith Carradine and Sandra Will have any children?

Context: [Q: Who proposed that atoms are the basic units of matter? A: John Dalton proposed that each chemical
element is composed of atoms of a single, unique type, and they can combine to form more complex structures
called chemical compounds.]
Question: How did the proposal come about?
Rewrite: How did John Dalton's proposal that each chemical element is composed of atoms of a single unique
type, and they can combine to form more complex structures called chemical compounds come about?

Context: [Q: What is it called when two liquids separate? A: Decantation is a process for the separation of
mixtures of immiscible liquids or of a liquid and a solid mixture such as a suspension. Q: How does the
separation occur? A: The layer closer to the top of the container-the less dense of the two liquids, or the
liquid from which the precipitate or sediment has settled out-is poured off.]
Question: Then what happens?
Rewrite: Then what happens after the layer closer to the top of the container is poured off with decantation?

Context: {context}
Question: {query}
Rewrite:
```

Figure 10: Prompt used for Query Rewriting.

frequency. For GTR[4], the maximum token length is set to 384 for both the reformulated query and passage.

Both sparse and dense retrieval systems retrieve the top-100 relevant passages per query, and the aforementioned metrics are computed using *pytrec-eval* (Van Gysel & de Rijke, 2018).

**Details of Response Generation.** We employ Llama-3.1-8b-instruct (Dubey et al., 2024) as the response generator. The top-4 most relevant passages, retrieved using BM25, are appended to the original query. The input query to BM25 is obtained by applying various CQR methods.

**Candidate Query Generation.** To generate diverse candidate queries, we build on prior studies (Yoon et al., 2024; Lai et al., 2025). Specifically, we utilize the gpt-3.5-turbo-0125 (OpenAI, 2022) via the OpenAI API[5] to transform user queries in conversational datasets into diverse candidate queries. The model is configured with a temperature of 0.8 and a top-$p$ value of 0.8 to promote diversity, with a maximum token limit set to 2560.

We adopt two prompting strategies: Question Rewriting and Query Expansion. Question Rewriting generates 12 candidate queries, whereas Query Expansion produces 3 additional candidates by applying Llama3.1-8b-instruct to the outputs of Question Rewriting. Both strategies are applied consistently across all datasets. The prompt templates for Question Rewriting (adapted from (Ye et al., 2023)) and Query Expansion (adapted from (Yoon et al., 2024)) are illustrated in Figure 10 and Figure 11, respectively.

# F  COMPLETE EXPERIMENT RESULTS

## F.1  EXTENDED RESULTS: QRECC

Table 13 extends the results of Table 2 by additionally using the QReCC dataset as the target dataset.

---

[4]https://huggingface.co/sentence-transformers/gtr-t5-large
[5]https://platform.openai.com/docs/models/gpt-3-5-turbo

```
Given a question, please answer the question in a sentence. The answer should be as informative as possible.

Question: and by whom was the game the last of us established?
Answer: Andy Gavin and Jason Rubin. Naughty Dog, LLC (formerly JAM Software, Inc.) is an American first-party
video game developer based in Santa Monica, California. Founded by Andy Gavin and Jason Rubin in 1984 as an
independent developer.

Question: is chelsea a club?
Answer: Yes, chelsea is an English professional football club.

Question: is call me by your name a movie?
Answer: Yes, based on a book of the same name. Call Me by Your Name is a 2017 coming-of- age romantic drama
film directed by Luca Guadagnino. Its screenplay, by James Ivory, who also co-produced, is based on the 2007
novel of the same name by Andr Aciman.

Question: where was ulysses s. grant from?
Answer: Hiram Ulysses Grant was born in Point Pleasant, Ohio, on April 27, 1822, to Jesse Root Grant, a tanner
and merchant, and Hannah Simpson Grant.

Question: {rewritten query}
Answer:
```

Figure 11: Prompt used for Query Expansion.

The results further highlight the importance of reference-free preference optimization by demonstrating that Upper Bound and RetPo, which leverage preference optimization, exhibit divergent performance depending on the availability of reference passages from the target dataset. Notably, DUALREFORM establishes itself as an effective approach for *reference-free* preference optimization, outperforming the baseline methods and achieving performance levels comparable to Upper Bound.

Table 13: Retrieval performance comparison of DUALREFORM against representative CQR baselines on the target dataset, QReCC. The best results (excluding Upper Bound) are highlighted in bold.

| Target Dataset | Query Reformulations | Sparse Retriever | | | | Dense Retriever | | | |
|---|---|---|---|---|---|---|---|---|---|
| | | MRR | NDCG | R@5 | R@20 | MRR | NDCG | R@5 | R@20 |
| | Upper Bound | 50.32 | 46.94 | 58.97 | 78.96 | 51.63 | 49.11 | 63.13 | 79.64 |
| | LLM-IQR | 41.82 | 38.88 | 52.58 | 71.95 | 48.09 | 45.25 | **62.13** | 80.24 |
| | HyDE-LLM | 42.26 | 39.21 | 52.93 | 72.44 | 48.20 | 45.46 | 61.97 | 79.85 |
| | LLM4CS-CoT | 47.51 | 44.25 | 56.64 | 78.96 | **48.53** | 45.49 | 58.54 | 79.64 |
| QReCC | T5QR | 32.19 | 29.17 | 40.18 | 61.94 | 41.21 | 38.18 | 54.63 | 73.45 |
| *SciConvQA* | ConvGQR | 32.49 | 29.66 | 41.36 | 59.47 | 36.86 | 34.16 | 48.94 | 67.72 |
| ↓ | HyDE-FT | 38.87 | 36.09 | 47.41 | 64.62 | 41.23 | 37.92 | 52.55 | 68.76 |
| *QReCC* | RetPo | 39.22 | 36.49 | 48.76 | 65.07 | 45.44 | 42.73 | 59.51 | 78.32 |
| | **DUALREFORM** | **48.40** | **45.30** | **58.91** | **77.90** | 47.58 | **46.98** | 59.60 | **80.33** |

## F.2 EXTENDED RESULTS: EFFECT OF PSEUDO REFERENCE REFINEMENT VIA CQR

Table 14 extends the results of Table 3 by additionally using other evaluation metrics, NDCG@3 and Recall@20, in order to offer a more comprehensive assessment of retrieval performance.

**Comparison with the Substantially Larger Backbone Language Model, Llama3.1-70b-inst.** In the Llama and Llama+ICL variants, we replace their backbones with a larger backbone, *Llama3.1-70b-inst*, whereas DUALREFORM keeps using *Llama3.1-8b-inst* as the backbone for its CQR model. As shown in Table 15, DUALREFORM achieves comparable or marginally superior performance despite using a substantially smaller language model as its backbone. This finding demonstrates that CQR can effectively substitute widely adopted LLMs for response refinement without introducing additional computational overhead.

**Demonstrations for Llama+ICL.** *Llama+ICL* differs from *Llama* by utilizing in-context demonstrations, as shown in Figure 12.

Table 14: Comparison of response refinement methods, evaluated using the sparse retriever. The highest values are emphasized in bold.

| Data | Refine Methods | Pseudo Ref. Acc. | | Retrieval Acc. | |
|---|---|---|---|---|---|
| | | NDCG | R@20 | NDCG | R@20 |
| SciConvQA | Llama | 35.62 | 59.37 | 16.53 | 34.03 |
| | Llama+ICL | 43.99 | 66.00 | 17.56 | 42.32 |
| | DUALREFORM | **49.36** | **71.27** | **18.88** | **42.78** |
| TopiOCQA | Llama | 35.42 | 50.40 | 22.98 | 53.62 |
| | Llama+ICL | 50.24 | 67.86 | 25.76 | 54.02 |
| | DUALREFORM | **56.23** | **76.82** | **26.57** | **59.03** |

Table 15: Comparison with the substantially larger Llama3.1-70b-inst backbone. Both Llama and Llama+ICL use Llama3.1-70b-inst, while DUALREFORM employs the smaller Llama3.1-8b-inst for CQR. The highest values are emphasized in bold.

| Data | Backbone Models | Refine Methods | Pseudo Reference Accuracy | | | |
|---|---|---|---|---|---|---|
| | | | MRR | NDCG | R@5 | R@20 |
| SciConvQA | Llama3.1-70B-inst | Llama | 47.61 | 46.91 | 57.14 | 68.28 |
| | | Llama+ICL | 49.64 | 48.87 | 58.46 | 69.90 |
| | Llama3.1-8B-inst | DUALREFORM | **50.05** | **49.36** | **59.75** | **71.27** |
| TopiOCQA | Llama3.1-70B-inst | Llama | 51.00 | 50.51 | 60.74 | 71.09 |
| | | Llama+ICL | 55.38 | 55.10 | 64.70 | 73.74 |
| | Llama3.1-8B-inst | DUALREFORM | **56.50** | **56.23** | **66.79** | **76.82** |

```
Context: [Q: When was Born to Fly released? A: Sara Evans's third studio album, Born to Fly, was released on
October 10, 2000.]
Explain: The last answer provides the release date of Sara Evans's third studio album, 'Born to Fly,' which
was October 10, 2000.

Context: [Q: When was Keith Carradine born? A: Keith Ian Carradine was born August 8, 1949. Q: Is he married?
A: Keith Carradine married Sandra Will on February 6, 1982.]
Explain: The last answer indicates that Keith Carradine married Sandra Will on February 6, 1982. In the
context of the overall conversation, the focus shifts from Keith Carradine's birth date to his marital status,
specifically addressing whether he is married.

Context: [Q: I've been curious about studies improving livestock productivity; is there a specific goal that
they often focus on? A: Undoubtedly, enhancing the efficiency of livestock breeds while minimizing their
ecological footprint are important goals in animal farming. Q: You mentioned improving certain efficiencies.
Which one takes up most of the cost in this setting? A: In animal agriculture, about 70% of the total
production expenses are attributed to feed costs.]
Explain: The last answer in this conversation states that in animal agriculture, about 70% of the total
production expenses are attributed to feed costs. This means that the majority of the expenses incurred in
livestock production are related to providing feed for the animals.
```

Figure 12: Demonstrations used for the Llama+ICL variant.

## F.3 EXTENDED RESULTS: EFFECT OF ITERATIVE REFINEMENT FOR PSEUDO REFERENCE

Table 16 extends the results of Table 4 by additionally using other evaluation metrics, NDCG@3 and Recall@20. These results again demonstrate that DUALREFORM benefits from the iterative refinement process of pseudo reference passages.

To gain a clearer view of the overall refinement dynamics, we further analyze how performance evolves as the number of refinement steps increases. Figure 13 depicts the evolution of both pseudo reference accuracy and retrieval accuracy when the number of iterative optimization steps is increased to five. The gains in retrieval accuracy largely converge around the third update, with subsequent iterations yielding only marginal improvements, which is consistent with prior observations on the saturation behavior of self-training methods (Xie et al., 2020). In contrast, pseudo reference accuracy continues to increase beyond the third update, but these improvements do not translate into higher retrieval accuracy. A plausible explanation is that repeated training on essentially the same data

Table 16: Effect of iterative optimization within DUALREFORM, evaluated using the sparse retriever. The highest values are emphasized in bold.

| Data | Pseudo Ref. Updates | Pseudo Ref. Acc. | | Retrieval Acc. | |
|---|---|---|---|---|---|
| | | NDCG | R@20 | NDCG | R@20 |
| SciConvQA | 1 | 37.48 | 58.98 | 15.50 | 40.23 |
| | 2 | 45.42 | 65.88 | 17.96 | 40.38 |
| | 3 | **49.36** | **71.27** | **18.88** | **42.78** |
| TopiOCQA | 1 | 38.82 | 50.75 | 23.66 | 52.11 |
| | 2 | 54.86 | 75.28 | 26.42 | 57.25 |
| | 3 | **56.23** | **76.82** | **26.57** | **59.03** |

distribution induces overfitting in the CQR model: while such overfitting improves training-side metrics (i.e., pseudo reference accuracy), it does not enhance test-side performance, leading to a plateau in retrieval accuracy.

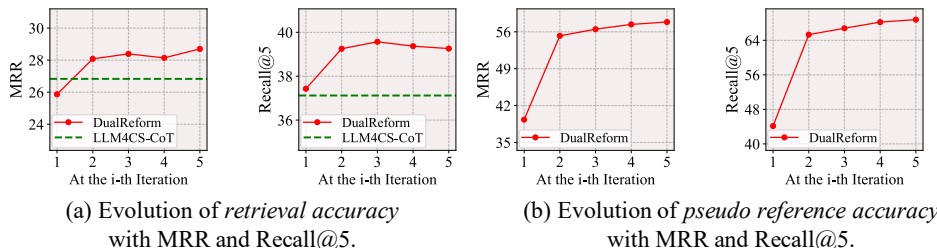

(a) Evolution of *retrieval accuracy*
with MRR and Recall@5.

(b) Evolution of *pseudo reference accuracy*
with MRR and Recall@5.

Figure 13: Evolution of retrieval and pseudo reference accuracy of DUALREFORM over iterative refinement steps: (a) shows *retrieval accuracy* at each refinement iteration, with LLM4CS-CoT plotted as a fixed dashed baseline; and (b) shows the corresponding *pseudo reference accuracy*.

## F.4 EFFECT OF QUERY-FORMING TEMPLATE

Table 17 extends the results of Table 5 by additionally using other evaluation metrics, NDCG@3 and Recall@20. These results further validate that the performance of the variants consistently declines compared to DUALREFORM, confirming the effectiveness of its query-forming template.

Table 17: Effect of the query-forming template on pseudo reference accuracy using the sparse retriever. The highest values are emphasized in bold.

| Variants | SciConvQA | | TopiOCQA | | Degrade |
|---|---|---|---|---|---|
| | NDCG | R@20 | NDCG | R@20 | |
| **No Template** | 44.56 | 65.70 | 54.23 | 73.45 | 6.88% |
| **No Response** | 46.59 | 68.72 | 49.93 | 71.71 | 7.35% |
| **No QE** | 42.84 | 62.75 | 53.94 | 72.57 | 9.72% |
| **No QR** | 45.90 | 66.33 | 46.58 | 66.58 | 12.77% |
| **DUALREFORM** | 49.36 | 71.27 | 56.23 | 76.82 | - |

## F.5 MORE EXAMPLES OF REFINED RESPONSES

Figure 14 complements Figure 5, presenting the complete refined responses generated by various methods. Figure 15 illustrates the top attention weights assigned by DUALREFORM during response refinement, demonstrating its ability to correctly attend to the relevant context (e.g., "Marvel Studios") and the target response (e.g., "Iron Man, X-Men, Spider-Man"). More examples for other conversations are provided in Figures 16–18.

| Method | Refined Responses $\tilde{a}_9$ | Conversation |
|--------|--------------------------------|--------------|
| **Reference Passage** | Marvel Studios Introduction Marvel Studios, LLC (originally known as Marvel Films from 1993 to 1996) is an American film and television studio that is a subsidiary of The Walt Disney Studios. Marvel Studios is known for the production of the Marvel Cinematic Universe films, based on characters that appear in Marvel Comics publications. Marvel Studios has released 23 films since 2008 within the Marvel Cinematic Universe, from "Iron Man" (2008) to "Spider-Man: Far From Home" (2019). These films all share continuity with each other, along with the One-Shots produced by the studio and the television shows produced by Marvel Television. In addition to the Marvel Cinematic Universe, Marvel Studios was also involved with the production of other Marvel-character film franchises that have exceeded $1 billion in North American box office revenue: the X-Men and Spider-Man multi-film franchises. | **< Conv. History >** $q_1$: who plays the general in the incredible hulk? $a_1$: Thaddeus "Thunderbolt". $q_2$: when was the movie released? $a_2$: June 13, 2008. $q_3$: what is the storyline? $a_3$: Bruce Banner becomes the Hulk as an unwitting pawn in a military scheme to reinvigorate the "Super-Soldier" program through gamma radiation. Banner goes on the run from the military while attempting to cure himself of the Hulk. $q_4$: who are the other characters? $a_4$: General Thaddeus "Thunderbolt" Ross, Betty, Dr. Samuel Sterns. $q_5$: who portrayed the protagonist of the movie? $a_5$: Bruce Banner. $q_6$: where did the filming take place? $a_6$: Toronto. $q_7$: who did the soundtrack? $a_7$: Craig Armstrong $q_8$: which production company is associated with it? $a_8$: Marvel Studios $q_9$: can you name any other movies they produced? |
| Llama-ICL | The last answer in this conversation lists several production-related details about the movie "The Incredible Hulk". It names the production company associated with the movie as Marvel Studios and mentions a few other movies produced by the same company, which are Iron Man, X-Men, and Spider-Man. | |
| No Template | Can you name other movies produced by Marvel Studios? Can you name other movies produced by Marvel Studios? … **(degenerated)** | |
| No Response | Can you clearly state the main points of the last answer, contextualizing them and resolving coreferences? The main points of the last answer are: Marvel Studios produced the movie The Incredible Hulk, which was released on June 13, 2008, and had a storyline where Bruce Banner becomes the Hulk as an unwitting pawn in a military scheme to reinvigorate the "Super-Soldier" program through gamma radiation. He goes on the run from the military while attempting to cure himself of the Hulk. The movie was filmed in Toronto, and the soundtrack was composed by Craig Armstrong. Marvel Studios is associated with the production of the movie and has also produced other notable films like Iron Man, X-Men, and Spider-Man. | **< Target Response >** $a_9$: Iron Man, X-Men, Spider-Man |
| **DualReform** | Can you clearly state the main points of the movies produced by Marvel Studios, contextualizing them and resolving coreferences? The main points of the movies produced by Marvel Studios include Iron Man, X-Men, and Spider-Man, which are three popular superhero films that have been well-received by audiences and critics alike. Iron Man is a film about a billionaire inventor who becomes a superhero, X-Men is a film about a group of mutants with unique powers, and Spider-Man is a film about a young man who gains powers after being bitten by a spider. All three films have been successful and have contributed to the Marvel Cinematic Universe. | |

Figure 14: Examples of refined responses generated by different methods on TopiOCQA. Fragments strongly aligned with the reference passage are highlighted in blue, while fragments with weaker connections (e.g., off-topic elements referring to previous conversation topics) are marked in red.

Context: [Q: who plays the general in the incredible hulk A: Thaddeus "Thunderbolt" Q: when was the movie released? A: June 13, 2008 Q: what is the storyline? A: Bruce Banner becomes the Hulk as an unwitting pawn in a military scheme to reinvigorate the "Super-Soldier" program through gamma radiation. Banner goes on the run from the military while attempting to cure himself of the Hulk. Q: who are the other characters ? A: General Thaddeus "Thunderbolt" Ross, Betty, Dr. Samuel Sterns Q: who portrayed the protagonist of the movie? A: Bruce Banner Q: where did the filming take place? A: Toronto Q: who did the soundtrack ? A: Craig Armstrong Q: which production company is associated with it? A: Marvel Studios Q: can you name any other movies they produced?] Question: Can you clearly state the main points of the last answer ('Iron Man, X-Men, Spider-Man'), contextualizing them and resolving coreferences?

Figure 15: Top attention weights assigned by DualReform during response refinement for the conversation in Figure 5, with high-weight regions highlighted in red.

| Method | Refined Responses $\tilde{a}_{10}$ | Conversation |
|---|---|---|
| **Reference Passage** | Atari, Inc. was an American video game developer and home computer company founded in 1972 by Nolan Bushnell and Ted Dabney. Primarily responsible for the formation of the video arcade and modern video game industry, the company assets were split in 1984 as a direct result of the video game crash of 1983, and the company was renamed first Atari Games, Inc. and then Atari Holdings, Inc. After 1985, when its coin-operated games division was sold to Namco and its arcade operations division was shut down, it remained a non-operating subsidiary of Warner Communications and its successor company, Time Warner, until it was merged into the parent company in 1992. | **< Conversation History >** $q_1$: when did the apple iphone 4s come out? $a_1$: October 14, 2011. $q_2$: which company manufactures this smartphone? $a_2$: Apple Inc. $q_3$: anything significant about the software used in this phone? $a_3$: It debuted with iOS 5, the fifth major version of iOS, Apple's mobile operating system, that introduced features including iCloud, iMessage, Notification Center, Reminders, and Twitter integration. $q_4$: was the product pulled into any controversy? $a_4$: Yes, on December 22, 2015, Apple faced a class action lawsuit for crippling the iPhone 4S with the iOS 9 update with slow and buggy software or to pay hundreds of dollars on a new iPhone. $q_5$: explain the notification feature mentioned above? $a_5$: Notification Center is a feature in iOS and macOS that provides an overview of alerts from applications, it displays notifications until the user completes an associated action, rather than requiring instant resolution. $q_6$: who owns the company that manufactures the phone? $a_6$: UNANSWERABLE $q_7$: who were the founders? $a_7$: Steve Jobs, Steve Wozniak, and Ronald Wayne. $q_8$: which university did the first person attend? $a_8$: Jobs enrolled at Reed College $q_9$: where did he have his first job? $a_9$: Atari, Inc. $q_{10}$: when was this founded? 

 **< Target Response >** $a_{10}$: 1972. |
| Llama+ICL | The last answer in this conversation states that the feature mentioned is Notification Center, which is a feature in iOS and macOS that provides an overview of alerts from applications, displaying notifications until the user completes an associated action, rather than requiring instant resolution. | |
| No Template | When was Atari, Inc. founded in 1972? Atari, Inc. was founded in 1972. | |
| No Response | Can you clearly state the main points of the last answer, contextualizing them and resolving coreferences? The main points of the last answer are: Steve Jobs, Steve Wozniak, and Ronald Wayne were the founders of Apple Inc., the company that manufactures the iPhone 4S. Steve Jobs enrolled at Reed College, and he had his first job at Atari, Inc., which was founded in 1972. The iPhone 4S was released on October 14, 2011, and it debuted with iOS 5, the fifth major version of iOS, which introduced features including iCloud, iMessage, Notification Center, Reminders, and Twitter integration. Apple faced a class action lawsuit in December 2015 for crippling the iPhone 4S with slow and buggy software. Notification Center is a feature in iOS and macOS that provides an overview of alerts from applications, displaying notifications until the user completes an associated action. | |
| **DUALREFORM** | Can you clearly state the main points of the founding of Atari, Inc. in 1972, contextualizing them and resolving coreferences? Atari, Inc. was founded in 1972 by Nolan Bushnell and Ted Dabney, and it is widely recognized as a pioneering company in the video game industry. The company was initially called Atari, Inc. and was later renamed to Atari Corporation in 1984. Atari, Inc. is known for its iconic games such as Pong, Asteroids, and Centipede, and it played a significant role in popularizing video games in the 1970s and 1980s. | |

Figure 16: Examples of refined responses generated by different methods on TopiOCQA. Fragments strongly aligned with the reference passage are highlighted in blue, while fragments with weaker connections (e.g., off-topic elements referring to previous conversation topics) are marked in red.

| Method | Refined Responses $\tilde{a}_{10}$ | Conversation |
|---|---|---|
| **Reference Passage** | The underlying bedrock or limestone of the Everglades basin affects the "hydroperiod", or how long an area within the region stays flooded throughout the year. Longer hydroperiods are possible in areas that were submerged beneath seawater for longer periods of time, while the geology of Florida was forming. More water is held within the porous ooids and limestone than older types of rock that spent more time above sea level. A hydroperiod of ten months or more fosters growth of sawgrass, whereas a shorter hydroperiod of six months or less promotes beds of periphyton, a growth of algae and other microscopic organisms. | **< Conversation History >** $q_1$: what is the longest river in the everglades? $a_1$: UNANSWERABLE $q_2$: in which state of the us is this region located? $a_2$: Florida. $q_3$: what is the type of ecosystem it has in this mentioned region of this state? $a_3$: It is a natural region of tropical wetlands, comprising the southern half of a large drainage basin within the Neotropical realm. |
| Llama+ICL | The last answer indicates that the rocks found in the region where the mentioned plants are found in the state of Florida are bedrock or limestone. This answer provides a piece of information about the geological characteristics of the area. | $q_4$: name some plants typically found in this place. $a_4$: Sawgrass prairies, bladderwort, waterlily etc. $q_5$: is there a botanical name of the second one? |
| No Template | Bedrock or limestone rocks are found in the region where these plants are found in the state of Florida. | $a_5$: Yes, "Utricularia" $q_6$: is this a carnivorous plant? $a_6$: Yes, it captures small organisms by means of bladder-like traps. |
| No Response | Can you clearly state the main points of the fact that certain plants in the Everglades region of Florida have self-pollinating flowers and may also produce open, insect-pollinated flowers elsewhere or at a different time of year, and that the region is characterized by the presence of bedrock or limestone rocks? The Everglades region in Florida is characterized by the presence of plants that have self-pollinating flowers and may also produce open, insect-pollinated flowers elsewhere or at a different time of year, and the region is also marked by the presence of bedrock or limestone rocks. The plants in this region, such as sawgrass prairies, bladderwort, and waterlily, have unique features like significantly greater respiration rates and complex energy-dependent traps, and certain species may produce closed, self-pollinating flowers in particular seasons, while others may produce open, insect-pollinated flowers elsewhere or at a different time of year. The Everglades region is a natural region of tropical wetlands, comprising the southern half of a large drainage basin within the Neotropical realm, and is characterized by the presence of bedrock or limestone rocks, which are found in the region where these plants are found. | $q_7$: what do they feed on? $a_7$: Protozoa and rotifers swimming in water-saturated soil. $q_8$: what is the peculiar feature of their respiration? $a_8$: They have significantly greater respiration rates than most vegetative tissue, primarily due to their complex energy-dependent traps. $q_9$: how does reproduction take place in these plants? $a_9$: Certain plants in particular seasons might produce closed, self-pollinating ("cleistogamous") flowers; but the same plant or species might produce open, insect-pollinated flowers elsewhere or at a different time of year, and with no obvious pattern. |
| **DUALREFORM** | Can you clearly state the main points of the answer 'Bedrock or limestone' in the context of the Everglades region in Florida, and resolve any coreferences or ambiguities? The main points of the answer 'Bedrock or limestone' in the context of the Everglades region in Florida are that the region is characterized by bedrock or limestone, which are the primary geological formations found in the area. These formations provide a foundation for the unique ecosystem of the Everglades, supporting the growth of plants such as sawgrass, bladderwort, and waterlily, and creating a habitat for a diverse range of wildlife. | $q_{10}$: what rocks are found in that us state in the region where these plants are found? **< Target Response >** $a_{10}$: Bedrock or limestone. |

Figure 17: Examples of refined responses generated by different methods on TopiOCQA. Fragments strongly aligned with the reference passage are highlighted in blue, while fragments with weaker connections (e.g., off-topic elements referring to previous conversation topics) are marked in red.

| Method | Refined Responses $\tilde{a}_9$ | Conversation |
|---|---|---|
| **Reference Passage** | "A Star Is Born", Enigma, and "Chromatica" In March 2018, Gaga supported the March for Our Lives gun-control rally in Washington, D.C., and released a cover of Elton John's "Your Song" for his tribute album "Revamp". Later that year, she starred as struggling singer Ally in Bradley Cooper's musical romantic drama "A Star Is Born", a remake of the 1937 film of the same name. The film follows Ally's relationship with singer Jackson Maine (played by Cooper), which becomes strained after her career begins to overshadow his. Cooper approached Gaga after seeing her perform at a cancer research fundraiser; an admirer of Cooper's work, Gaga agreed to the project due to its portrayal of addiction and depression. | **< Conversation History >** $q_1$: who is lady gaga? $a_1$: Stefani Joanne Angelin. Germanotta, known professionally as Lady Gaga, is an American singer, songwriter and actress. $q_2$: where was she born? $a_2$: Manhattan, New York City. $q_3$: when? $a_3$: March 28, 1986. $q_4$: can you please name some of her songs? $a_4$: Bad Romance", "Telephone", and "Alejandro". $q_5$: describe her music style? $a_5$: Electronic rock and techno. $q_6$: describe her music style? $a_6$: Electronic rock and techno. $q_7$: can you please name some of her albums? $a_7$: "The Fame", "Born This Way", "Artpop". $q_8$: has she won any awards for her music? $a_8$: Yes, 11 Grammy Awards, an Academy Award, a BAFTA Award, three Brit Awards, two Golden Globe Awards, eighteen MTV Video Music Awards, various Guinness World Record. $q_9$: has she acted in movies? 

 **< Target Response >** $a_9$: Yes, "A Star Is Born". |
| Llama+ICL | In the context of the overall conversation, the last answer mentions Lady Gaga's acting career, specifically that she has acted in the movie "A Star Is Born". | |
| No Template | Has Lady Gaga acted in movies, including "A Star Is Born"? Yes, Lady Gaga has acted in movies, including "A Star Is Born". | |
| No Response | Can you clearly state the main points of Lady Gaga's awards and acting career? Lady Gaga has won 11 Grammy Awards, an Academy Award, a BAFTA Award, three Brit Awards, two Golden Globe Awards, eighteen MTV Video Music Awards, and various Guinness World Records for her music. Additionally, she has acted in the movie "A Star Is Born". | |
| **DUALREFORM** | Can you clearly state the main points of the answer "Yes, "A Star Is Born" in relation to Lady Gaga's acting career? Lady Gaga has acted in the movie "A Star Is Born." The movie "A Star Is Born" is a musical romantic drama film that stars Lady Gaga and Bradley Cooper. Lady Gaga's performance in the movie earned her an Academy Award for Best Original Song for "Shallow." | |

Figure 18: Examples of refined responses generated by different methods on TopiOCQA. Fragments strongly aligned with the reference passage are highlighted in blue, while fragments with weaker connections (e.g., off-topic elements referring to previous conversation topics) are marked in red.

## F.6 GENERATION ACCURACY ON TOPIOCQA

Table 18 extends the results of Table 7 by additionally using the TopiOCQA dataset. These results show that DUALREFORM consistently outperforms the baselines by achieving higher generation accuracy through accurate passage retrieval across diverse conversational domains.

Table 18: Response generation accuracy with passages retrieved via different CQR methods on TopiOCQA. The highest values are emphasized in bold.

| CQR Methods | Generation Accuracy | | | |
|---|---|---|---|---|
| | LLMEval | ROUGE-1 | ROUGE-L | BertScore |
| LLM-IQR | 25.34 | 20.36 | 19.09 | 84.32 |
| HyDE-LLM | 27.36 | 24.83 | 23.30 | 85.28 |
| LLM4CS-CoT | 33.78 | 29.20 | 27.78 | 85.88 |
| T5QR | 18.92 | 17.07 | 16.50 | 83.67 |
| ConvGQR | 21.62 | 22.02 | 20.89 | 84.84 |
| HyDE-FT | 15.98 | 15.37 | 14.72 | 84.01 |
| RetPo | 33.11 | 26.01 | 24.39 | 85.74 |
| **DUALREFORM** | **35.81** | **31.63** | **30.10** | **86.48** |

## F.7 GENERALIZATION TO AN ALTERNATIVE DENSE RETRIEVER

Table 19 reports the performance of all CQR models under ANCE (Xiong et al., 2021), another widely used dense retriever, on SciConvQA and TopiOCQA. Overall, we observe that DUALREFORM continues to outperform the CQR baselines on both datasets, indicating that the gains of DUALRE-FORM are not an artifact of particular retrieval systems. Interestingly, ANCE achieves noticeably

lower scores than GTR on SciConvQA (around a 10% drop across MRR, nDCG, and R@5), which we attribute to GTR, as a more recent model, being better adapted to the specialized scientific domain covered by SciConvQA. In contrast, on the more general TopiOCQA dataset, the average degradation of ANCE with respect to GTR is much smaller (about 1–5% across the three metrics), suggesting that both retrievers are sufficiently trained for broad, non-specialized topics.

Table 19: Comparison of all CQR methods under two dense retrievers. with the last row in each block showing the average degradation of ANCE w.r.t. GTR, computed as $\frac{(ANCE-GTR)}{GTR}100\%$.

| Dataset | CQR Methods | MRR | | NDCG | | R@5 | |
|---------|-------------|-----|-----|------|-----|-----|-----|
| | | GTR | ANCE | GTR | ANCE | GTR | ANCE |
| SciConvQA | HyDE-LLM | $16.70_{\pm0.69}$ | $14.99_{\pm0.17}$ | $15.38_{\pm0.66}$ | $14.04_{\pm0.33}$ | $23.28_{\pm0.70}$ | $20.93_{\pm0.34}$ |
| | LLM4CS-CoT | $18.25_{\pm0.22}$ | $16.68_{\pm0.10}$ | $16.75_{\pm0.18}$ | $15.56_{\pm0.10}$ | $24.48_{\pm0.22}$ | $22.33_{\pm0.37}$ |
| | T5QR | $14.51_{\pm0.74}$ | $11.18_{\pm0.08}$ | $13.44_{\pm0.77}$ | $10.20_{\pm0.09}$ | $19.42_{\pm1.12}$ | $14.72_{\pm0.14}$ |
| | ConvGQR | $14.67_{\pm0.10}$ | $14.14_{\pm0.09}$ | $13.72_{\pm0.16}$ | $13.14_{\pm0.09}$ | $20.03_{\pm0.08}$ | $18.87_{\pm0.11}$ |
| | HyDE-FT | $12.16_{\pm1.03}$ | $10.95_{\pm1.30}$ | $11.09_{\pm1.15}$ | $10.01_{\pm1.34}$ | $16.63_{\pm1.90}$ | $15.02_{\pm1.33}$ |
| | RetPO | $17.95_{\pm0.08}$ | $16.30_{\pm0.25}$ | $16.74_{\pm0.08}$ | $15.29_{\pm0.21}$ | $24.31_{\pm0.36}$ | $21.46_{\pm0.25}$ |
| | DUALREFORM | $23.53_{\pm0.09}$ | $20.29_{\pm0.18}$ | $22.04_{\pm0.23}$ | $19.67_{\pm0.09}$ | $31.19_{\pm0.18}$ | $27.93_{\pm0.29}$ |
| **Average Degradation (%)** | | – | $-11.19$ | – | $-10.47$ | – | $-11.53$ |
| TopiOCQA | HyDE-LLM | $33.39_{\pm1.48}$ | $31.83_{\pm1.11}$ | $32.02_{\pm1.34}$ | $30.54_{\pm1.39}$ | $45.54_{\pm0.05}$ | $42.39_{\pm0.65}$ |
| | LLM4CS-CoT | $37.55_{\pm0.75}$ | $38.91_{\pm1.04}$ | $35.92_{\pm1.04}$ | $38.18_{\pm1.03}$ | $52.45_{\pm0.35}$ | $51.95_{\pm0.61}$ |
| | T5QR | $28.36_{\pm0.07}$ | $23.25_{\pm0.07}$ | $27.42_{\pm0.05}$ | $22.16_{\pm0.10}$ | $39.78_{\pm0.07}$ | $33.62_{\pm0.19}$ |
| | ConvGQR | $22.58_{\pm0.14}$ | $24.83_{\pm0.35}$ | $21.33_{\pm0.19}$ | $23.61_{\pm0.52}$ | $33.60_{\pm0.07}$ | $35.41_{\pm0.24}$ |
| | HyDE-FT | $19.28_{\pm0.63}$ | $17.80_{\pm1.13}$ | $17.80_{\pm0.80}$ | $16.83_{\pm0.83}$ | $27.01_{\pm0.88}$ | $23.86_{\pm0.43}$ |
| | RetPO | $35.67_{\pm0.05}$ | $35.29_{\pm0.23}$ | $34.28_{\pm0.04}$ | $34.35_{\pm0.30}$ | $49.99_{\pm0.31}$ | $48.55_{\pm0.40}$ |
| | DUALREFORM | $40.14_{\pm0.21}$ | $39.11_{\pm0.60}$ | $38.33_{\pm0.28}$ | $38.38_{\pm0.72}$ | $53.78_{\pm0.58}$ | $51.90_{\pm0.84}$ |
| **Average Degradation (%)** | | – | $-2.92$ | – | $-1.71$ | – | $-5.14$ |

### F.8 COMPONENT-LEVEL ABLATION OF DUALREFORM

To better understand which components of DUALREFORM are responsible for the observed gains, Table 20 provides a per-component breakdown for generating pseudo reference passages by progressively activating (●): (i) *retrieval of pseudo reference passages* that enables our reference-free approach, (ii) *LLM-based response refinement*, and (iii) *CQR-based response refinement*. Overall, activating pseudo reference retrieval (Variant 1 → Variant 2) yields the largest performance increase, indicating that the reference-free approach itself is the primary driver of the gains. The subsequent response refinement stages, particularly the transition from Variant 3 to DUALREFORM (activating CQR refinement), provide further noticeable gains by improving the quality of pseudo reference inference via higher-quality responses.

Table 20: Component-level ablation of DUALREFORM. We progressively enable pseudo reference retrieval, LLM-based response refinement, and CQR-based response refinement (Variants 1–3), with DUALREFORM using all three components. Here, ○ and ● indicate that a component is disabled or enabled, respectively.

| | Main components | | | TopiOCQA | | SciConvQA | |
|---------|-----------|-----------|------------------|------|-----|------|-----|
| Variant | CQR refine | LLM refine | Pseudo ref. retr. | R@5 | MRR | R@5 | MRR |
| Variant 1 | ○ | ○ | ○ | 32.18 | 23.20 | 21.61 | 15.60 |
| Variant 2 | ○ | ○ | ● | 37.43 | 25.87 | 26.05 | 17.55 |
| Variant 3 | ○ | ● | ● | 37.79 | 27.92 | 26.77 | 19.02 |
| DUALREFORM | ● | ● | ● | **39.57** | **28.39** | **28.64** | **20.06** |

### F.9 PARAMETER SENSITIVITY ANALYSIS

We conduct the sensitivity analysis of DUALREFORM's hyperparameters, specifically the number of pseudo reference passages $k$ in Definition 4.5 and the regularization parameter $\beta$ of DPO in Eq. (10).

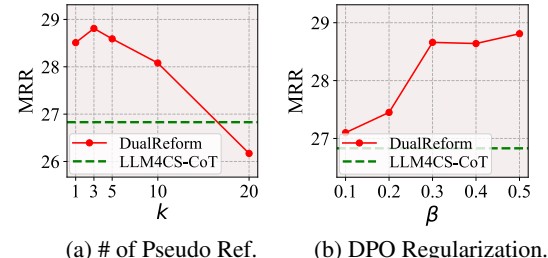

(a) # of Pseudo Ref.  (b) DPO Regularization.

Figure 19: Effects of the hyperparameters of DUALREFORM on MRR. The green dashed line denotes the performance of LLM4CS-CoT, the strongest baseline.

**Number of Pseudo Reference Passages** $k$. Figure 19(a) presents the impact of $k$ on retrieval performance. The parameter $k$ controls the number of top-ranked passages used as pseudo reference passages. Lower values selectively include only the most relevant passages, while higher values introduce additional but less relevant passages. In general, the retrieval accuracy stabilizes between 1 and 5, after which it declines, indicating a negative impact from less relevant passages beyond a specific threshold.

**DPO Regularization Parameter** $\beta$. Figure 19(b) presents retrieval accuracy across different values of $\beta \in \{0.1, 0.2, 0.3, 0.4, 0.5\}$, as guided by prior works (Rashul et al., 2023; Furuta et al., 2024). This parameter controls the trade-off between aligning the model with user preferences and retaining the behavior of the pre-trained model. Lower values prioritize the former, while higher values place greater emphasis on the latter. In general, increasing $\beta$ improves performance, with a plateau observed around 0.3 and a peak at 0.5. This trend is attributed to the ability of higher $\beta$ values to reduce overfitting to the initial pseudo reference passages, further enabling model refinements in DUALREFORM through iterative optimization.

