# OpenReview forum: "References Indeed Matter? Reference-Free Preference Optimization for Conversational Query Reformulation"
_ICLR.cc/2026/Conference — Submitted to ICLR 2026_

### Official Review · Reviewer_m8aF · 2025-10-27

**Soundness:** 3
**Presentation:** 3
**Contribution:** 3
**Rating:** 6
**Confidence:** 4

**Summary:**

This paper presents DualReform, a reference-free preference optimization framework for conversational query reformulation (CQR). The work addresses a practical limitation: existing preference optimization methods for CQR require reference passages that are rarely available in real-world conversational datasets. The authors propose generating pseudo reference passages from query-response pairs through (1) response-based inference and (2) response refinement via a dual-role CQR model. The term ``dual-role'' arises from using the CQR model for both query reformulation and response refinement, leveraging the inherent alignment between these two objectives. A key novelty is the theoretical demonstration that the dual-role configuration induces a smaller (tighter) upper bound on training error compared to the single-role configuration, as formalized through error bounds based on pseudo label denoising theory. Experiments on QReCC, TopiOCQA, and a new scientific dataset (SciConvQA) show that DualReform achieves 96.9-99.1\% of the retrieval accuracy obtainable with ground-truth references and outperforms the strongest baseline by 15.7\% on average across sparse and dense retrieval systems.

**Strengths:**

1. The authors provide complete proofs of the Lemmas in the paper, which enhances the theoretical rigor and reproducibility of the work.
2. The proposed framework, DualReform, outperforms the compared LLM-based, SFT-only, and reference-based CQR methods on the evaluated benchmarks (TopiOCQA and SciConvQA).
3. The authors introduce SciConvQA, a new benchmark for conversational query reformulation in specialized scientific domains. The dataset is constructed using scientific journal data provided by the Korea Institute of Science and Technology Information (KISTI), with conversations generated using GPT-4o followed by manual quality validation.
4. The authors provide code for their implementation.

**Weaknesses:**

1. The paper uses text-wrapped figures in several places, which sometimes affects readability and could be improved by repositioning these figures or reducing the use of text wrapping to enhance the flow and clarity of the paper.
2. The field of LLM-based CQR method is evolving rapidly. However, the most recent methods that the authors chose as baselines were published in 2023. The authors should include recent papers in the field, at least those published in 2024, to ensure the related work and comparisons reflect the current state-of-the-art. For example, the authors could refer to recent work such as Yunah Jang et al., “IterCQR: Iterative Conversational Query Reformulation with Retrieval Guidance,” NAACL 2024, and Fengran Mo et al., “CHIQ: Contextual History Enhancement for Improving Query Rewriting in Conversational Search,” EMNLP 2024.
3. The authors evaluate DualReform using only BM25 and GTR as retrieval systems. The work could be strengthened by extending the evaluation to other retrieval systems such as SPLADE and ANCE, which are commonly adopted in the research field of conversational retrieval.
4. The theoretical analysis relies on Assumption 1, which requires c-expansion and μ-separation properties of the data distribution. However, the authors do not verify whether the experimental datasets (QReCC, TopiOCQA, SciConvQA) actually satisfy these assumptions, nor do they provide the measured values of c and μ for these datasets. Please refer to the first point in the Questions section.

**Questions:**

Regarding line 105-112 in the methodology section, the authors derive training error bounds (Lemmas 1 and 2) under Assumption 1, which requires $c$-expansion and $\mu$-separation properties of the data distribution. Did the authors verify that the datasets used in the experiments (e.g., QReCC, TopiOCQA, SciConvQA) actually satisfy these assumptions? If so, what are the measured values of $c$ and $\mu$ for these datasets? Since the values of $c$ and $\mu$ determine whether the assumptions and the mathematical reasoning process hold, the authors should provide these values for each dataset.

---

> ### Author Response · Authors · 2025-11-21
>
> `W1. The paper uses text-wrapped figures in several places, which sometimes affects readability and could be improved by repositioning these figures or reducing the use of text wrapping to enhance the flow and clarity of the paper.`
>
> We apologize for any readability issues caused by text-wrapped figures, and sincerely thank you for your constructive feedback, including the insightful questions below. We will certainly reorganize all figures, tables, and algorithms when we prepare the camera-ready version.
>
> ---
>
> `W2. Could the authors include recent papers, at least those published in 2024, to ensure that the related work and comparisons reflect the current state-of-the-art?`
>
>
> We appreciate the reviewer’s suggestion and fully agree that incorporating recent work, at least from 2024, is important. Accordingly, we have extended our comparisons to include the **most recent CQR method AdaCQR [Lai et al., 2025]**, as shown in Table R11. AdaCQR, similar to RetPO [Yoon et al., 2024], leverages human-annotated reference passages to perform preference optimization, while further strengthening the preference signal by ensembling multiple retrievers.
>
> Table R11 compares DualReform with strong CQR baselines, including AdaCQR, on SciConvQA and TopiOCQA for both sparse and dense retrievers. Because AdaCQR (like RetPO) is explicitly designed to exploit reference passages, its performance in a **reference-free setting**—where no reference passages are available for the target dataset—remains close to RetPO and clearly below DualReform. This result further underscores both the **practical necessity** of our reference-free approach and the **effectiveness** of DualReform in such realistic scenarios.
>
>
> | Data      | Query Reformulations                  | Sparse MRR    | Sparse NDCG  |  Dense MRR    | Dense NDCG  |
> |-----------|----------------------------------------|---------------|--------------|--------------|--------------|
> | SciConvQA | Upper Bound                   | 20.89±0.30    | 18.91±0.58   |  23.73±0.70   | 22.49±0.80  |
> |  | LLM4CS-CoT                             | 16.53±0.18    | 15.29±0.27   |  18.25±0.22   | 16.75±0.18  |
> |  | RetPo                                  | 15.60±0.23    | 14.36±0.24   |  17.95±0.08   | 16.74±0.08  |
> |  | **AdaCQR**                                  | 15.97±0.31    | 13.97±0.34   |  20.14±0.17   | 18.83±0.19  |
> |  | **DualReform** (ours)                      | **20.06±0.19**    | **18.42±0.27**   |  **23.53±0.09**   | **22.04±0.23**  |
> | TopiOCQA  | Upper Bound                   | 29.19±0.30    | 27.19±0.22   |  40.58±0.22   | 39.15±0.25  |
> |   | LLM4CS-CoT                            | 26.81±0.66    | 25.40±0.38   |  37.55±0.75   | 35.92±1.04  |
> |   | RetPo                                 | 23.20±0.33    | 21.41±0.28   |  35.67±0.05   | 34.28±0.04  |
> |   | **AdaCQR**                                 | 24.78±0.63    | 22.60±0.78   |  36.45±0.50   | 35.20±0.56  |
> |   | **DualReform** (ours)                     | **28.39±0.39**    | **26.08±0.46**   |  **40.14±0.21**   | **38.33±0.28**  |
>
> *Table R11*: Retrieval accuracy on SciConvQA and TopiOCQA with representative CQR baselines, extended to include the **most recent method AdaCQR [Lai et al., 2025]** baseline for both sparse and dense retrievers.

---

> > ### Author Response · Authors · 2025-11-21
> >
> > `W3. The authors evaluate DualReform using only BM25 and GTR as retrieval systems. The work could be strengthened by extending the evaluation to other retrieval systems such as SPLADE and ANCE, which are commonly adopted in the research field of conversational retrieval.`
> >
> >
> > We are grateful for the opportunity to further assess the robustness of DualReform across different retrieval systems. To this end, we have additionally run dense-retrieval experiments with **ANCE** [Xiong et al., 2020] for all CQR methods, as shown in Table R12.
> >
> > Overall, we observe that the relative performance of ANCE and GTR [Ni et al., 2022] varies by dataset. On **SciConvQA**, ANCE achieves roughly 8–10% **lower** performance than GTR, which we hypothesize is due to GTR, as a more recent model, being better trained for the specialized scientific domain. In contrast, on the more general **TopiOCQA** dataset, GTR and ANCE perform **similarly**, suggesting that both retrievers are sufficiently trained for broad, non-specialized topics.
> >
> >
> > | Data | CQR  | MRR |  | nDCG |  | R@5 | |
> > |---------|------|-----|-----|------|------|-----|-----|
> > |         |      | GTR | **ANCE** | GTR  | **ANCE** | GTR | **ANCE** |
> > | SciConvQA| HyDE-LLM          | 16.70±0.69  | 14.99±0.17     | 15.38±0.66  | 14.04±0.33      | 23.28±0.70  | 20.93±0.34     |
> > |          | LLM4CS-CoT        | 18.25±0.22  | 16.68±0.10     | 16.75±0.18  | 15.56±0.10      | 24.48±0.22  | 22.33±0.37     |
> > |          | RetPO             | 17.95±0.08  | 16.30±0.25     | 16.74±0.08  | 15.29±0.21      | 24.31±0.36  | 21.46±0.25     |
> > |          | DualReform (ours) | 23.53±0.09  | 20.29±0.18     | 22.04±0.23  | 19.67±0.09      | 31.19±0.18  | 27.93±0.29     |
> > | -        | **average degradation (%)** |   -  | **-10.45**             |  -    |**-8.81**          |    -  | **-10.26**            |
> > | TopiOCQA | HyDE-LLM          | 33.39±1.48  | 31.83±1.11     | 32.02±1.34  | 30.54±1.39      | 45.54±0.05  | 42.39±0.65     |
> > |          | LLM4CS-CoT        | 37.55±0.75  | 38.91±1.04     | 35.92±1.04  | 38.18±1.03      | 52.45±0.35  | 51.95±0.61     |
> > |          | RetPO             | 35.67±0.05  | 35.29±0.23     | 34.28±0.04  | 34.35±0.30      | 49.99±0.31  | 48.55±0.40     |
> > |          | DualReform (ours) | 40.14±0.21  | 39.11±0.60     | 38.33±0.28  | 38.38±0.72      | 53.78±0.58  | 51.90±0.84     |
> > | -        | **average degradation (%)** |  -   | -1.17              |    -  | +0.50              |   -   | -3.56              |-
> >
> > *Table R12*: Comparison of two **dense retrievers** **(GTR vs. ANCE)**, with the last row in each block showing the **average degradation** of ANCE w.r.t. GTR, computed as (ANCE−GTR)/GTR×100%.

---

> > > ### Author Response · Authors · 2025-11-21
> > >
> > > `W4/Q1. Did the authors verify that the datasets used in the experiments satisfy Assumption 1 (c-expansion and µ-separation properties)?`
> > >
> > > We thank the reviewer for this careful question regarding Assumption 1 used in Lemma 4.1 [Wei et al., 2021]. These assumptions are defined at the level of the population distribution: the constants $c$ and $\mu$ are quantified over all measurable neighborhoods under the full population distribution, rather than over a finite empirical sample. Because we only have access to finite benchmarks drawn from this distribution, these population-level quantities are, in general, **not** directly identifiable from the data [Cai et al., 2021; Lang et al., 2022; Park et al., 2023; Deng et al., 2024; Mitsuzumi et al., 2024].
> > >
> > > Nevertheless, to make our theoretical analysis more concrete, we attempt to estimate $c$ and $\mu$ on our datasets under a **practically motivated relaxation** of the original conditions. Table R13 the resulting estimates. **All datasets satisfy $c>3$ with a relatively small $\mu$, which supports that they fall into the regime described by Assumption 1**. We hypothesize that the somewhat larger $\mu$ on TopiOCQA is due to its higher topical diversity and the resulting *sparsity* of the empirical sample distribution.
> > >
> > >
> > > | Dataset   | $c$       | $\mu$     |
> > > |----------|-------------|------------|
> > > | SciConvQA | 3.00010      | 0.00060     |
> > > | TopiOCQA | 3.00002    | 0.01932   |
> > > | QReCC | 3.00002    | 0.01568   |
> > >
> > > *Table R13*: Empirical estimates of the 𝑐-expansion and 𝜇-separation constants under our relaxed neighbor definition.
> > >
> > > Our relaxation proceeds as follows. We first need to define “neighbors” at the sample level. In the population-level definition, neighbors are examples that (i) share the same reference passage and (ii) lie within a radius $r$. In finite conversational datasets, however, examples with the same reference passage are much rarer than in the underlying population. We therefore relax **“same” to “similar”** reference passages: in SciConvQA, passages from the same article (which is split into multiple passages) are treated as *similar*; in TopiOCQA and QReCC, passages within the same topic are treated as *similar*, using the dataset’s topic labels for TopiOCQA and labels derived from clustering passage embeddings for QReCC. Meanwhile, the values of $c$ and $\mu$ vary with the choice of radius $r$, but $r$ is **not fixed** a priori in Lemma 4.1 [Wei et al., 2021]. Thus, we select the **smallest radius $r$ such that $c>3$**, and then verify that the resulting $\mu$ is sufficiently small. This procedure yields $r$=0.3278 for SciConvQA, 0.3414 for TopiOCQA, and 0.3419 for QReCC.
> > >
> > > *Implementation Details.* Distances between examples are computed in an embedding space obtained by encoding each example (current query plus the immediately preceding turn) using the sentence-transformer GTR-T5-large with a maximum token length of 384.
> > >
> > > ---
> > >
> > > [Ni et al., 2022] Large Dual Encoders Are Generalizable Retrievers. EMNLP, 2022
> > >
> > > [Xiong et al., 2020] Approximate nearest neighbor negative contrastive learning for dense text retrieval. arXiv, 2020
> > >
> > > [Yoon et al.,2024] Ask Optimal Questions: Aligning Large Language Models with Retriever’s Preference in Conversational Search. arXiv, 2024
> > >
> > > [Lai et al., 2025] AdaCQR: Enhancing Query Reformulation for Conversational Search via Sparse and Dense Retrieval Alignment. Coling, 2025
> > >
> > > [Wei et al., 2021] Theoretical Analysis of Self-Training with Deep Networks on Unlabeled Data. ICLR, 2021
> > >
> > > [Park et al., 2023] Robust Data Pruning under Label Noise via Maximizing Re-labeling Accuracy. NeurIPS, 2023
> > >
> > > [Mitsuzumi. et al, 2024] Understanding and Improving Source-free Domain Adaptation from a Theoretical Perspective. CVPR, 2024
> > >
> > > [Cai et al., 2021] A Theory of Label Propagation for Subpopulation Shift. ICML. 2021
> > >
> > > [Lang et al., 2022] Co-training Improves Prompt-based Learning for Large Language Models. ICML. 2022
> > >
> > > [Deng et al., 2024] Enhancing Large Vision Language Models with Self-Training on Image Comprehension. NeurIPS, 2024

---

> > > > ### Author Response · Authors · 2025-11-25
> > > > **Kindly Follow-Up**
> > > >
> > > > Dear Reviewer m8aF:
> > > >
> > > > We greatly appreciate your thoughtful feedback and the time devoted to this review, which substantially improved the quality of our work. If possible, we would be grateful for any additional insights at your convenience, as they would help us further improve this work. The authors would also be happy to discuss any aspects in more detail. Thank you very much.
> > > >
> > > > Sincerely,
> > > >
> > > > Authors

---

> > > > > ### Author Response · Authors · 2025-12-01
> > > > > **For Area Chair: Summary of Responses to Reviewer m8aF**
> > > > >
> > > > > The reviewer's requests focus on (i) the applicability of the theoretical assumptions in practice, (ii) the recency of baselines, and (iii) broader retriever coverage. In the rebuttal, we add targeted experiments and analyses that directly address the concerns, showing that the assumptions are empirically plausible, strengthening the baseline comparisons, and confirming robustness to the retriever choice (Tables R11–R13).
> > > > >
> > > > > | Comment | Concern                     | Response                                                                                                                                                                                                                                  |
> > > > > | ------- | --------------------------- | ----------------------------------------------------------------------------------------------------------------------------------------------------------------------------------------------------------------------------------------- |
> > > > > | Q1, W4   | Assumption verification     | **Additional results.** We estimate the assumption-related quantities on our datasets, indicating that the datasets **satisfy** the assumption. See **Table R13**. |
> > > > > | W2      | More recent baseline     | **Additional results.** We add the **most recent** strong baseline (AdaCQR) to our comparisons, which reconfirms DualReform's effectiveness. See **Table R11**. |
> > > > > | W3      | Dense retriever coverage          | **Additional results.** We conduct experiments with an **additional dense retriever** (ANCE), reconfirming the robustness of DualReform across retrievers. See **Table R12**.                                                                       |
> > > > >
> > > > > We believe this rebuttal addresses the reviewer's concerns, and the reviewer would therefore be inclined to increase his/her rating.

---

### Official Review · Reviewer_Kn6Q · 2025-10-29

**Soundness:** 3
**Presentation:** 3
**Contribution:** 2
**Rating:** 4
**Confidence:** 3

**Summary:**

This paper focuses on the Conversational Query Reformulation (CQR).  The authors find that current CQR methods are facing an issue, i.e.,  heavy reliance on reference passages for preference optimization, which are impractical to acquire in real-world scenarios.  To this end, the authors propose `DUALREFORM`, a reference-free preference optimization framework that generates pseudo reference passages from standard conversational datasets (containing only queries and responses) instead of relying on human-annotated references. DUALREFORM is an iterative framework, where the first step is the pseudo-reference generation and the next step is the preference optimization.
﻿
According to the experiments, DUALREFORM has achieved strong performance across three datasets (QReCC, TopiOCQA, and the novel specialized-domain SciConvQA): it reaches 90%+ of the retrieval accuracy of the reference-based method RetPo (Upper Bound).

**Strengths:**

1. The major part of this work is easy to follow.  The authors have given a nice preliminary study.
﻿
2. This work may be the first to achieve a reference-free preference optimization framework for CQR.
﻿
3. This work has provided a lot of theoretical theories to prove dual-role CQR reduces training error vs. single-role (LLM-only) refinement.
﻿
4. Achieves strong empirical performance: 1)  90%+ of reference-based accuracy,  2) 15.7% average improvement over SOTA reference-free baseline and 3) strong robustness across several datasets (domains).

**Weaknesses:**

1. Some statements are not very clear in this paper. I know that most of them may have been well validated in prior works, but the reader of paper may not familiar with them.  For example, it is hard to infer Lemmas 4.2 and 4.3 solely from the Assumption. Note that reading the appendix is not compulsory. Thus, the authors should give at least an easy but intuitive brief introduction in the main paper.
﻿
2.  I do not directly focus on this research field, so I have a concern about the definition of `reference-free`.  From my own perspective, this work is still  `reference-based` because, in step 2, this work still needs the reference, although this reference is a pseudo-reference. Thus,  compared to traditional reference-based methods,  the major difference is an automatic way to label the reference.
﻿
3. In Table 6 (Sec 5.4), the authors use some reference-aware metrics to evaluate the generated dialogues.  It would be better to use more reference-free metrics (just like LLMEval).

**Questions:**

1. Can you provide the performance with more iterations (i.e., Table 4, > 3).

---

> ### Author Response · Authors · 2025-11-21
>
> We sincerely appreciate the reviewer’s valuable time and feedback throughout the review process. We hope the clarifications in this rebuttal address your concerns.
>
>
> `Q1. Can you provide the performance with more iterations (i.e., Table 4, > 3)?`
>
> We appreciate the opportunity to clarify the performance trend of DualReform under additional iterations. To this end, we extend the number of pseudo reference refinement and CQR training steps to ***five*** iterations in total. For a clearer view of the effect of iteration, we refer the reviewer to a [`new figure`](https://drive.google.com/file/d/14-rr6374MkmyaAlxa3apOzkrii7gasJO/view?usp=sharing) that illustrates the evolution of pseudo reference and retrieval accuracy across iterations, at this [`🔗link`](https://drive.google.com/file/d/14-rr6374MkmyaAlxa3apOzkrii7gasJO/view?usp=sharing). In addition, Table R9 shows the corresponding results without accessing to the link for the reviewer’s convenience.
>
> Overall, **the gains in retrieval accuracy converge around the third update**, with subsequent iterations yielding only marginal or negligible improvements. This behavior is in line with prior observations in self-training approach [Xie et al., 2020], where the performance gain is saturated around the third iteration. By contrast, although **pseudo reference accuracy continues to rise** beyond the third update, these gains do **not** translate into increased retrieval accuracy. We hypothesize that this discrepancy arises from repeated training on the same distribution, causing the CQR model to become overfitted to the training data. While such overfitting improves training-side metrics (i.e., pseudo-reference accuracy), it does not enhance test-side performance, thereby resulting in a plateau in retrieval accuracy.
>
>
> |  | Retrieval Accuracy  |  | Pseudo Reference Accuracy  |  |
> |---------------------|-----------------|-----------------|--------------------|--------------------|
> | **Pseudo Reference Updates** |  MRR |  Recall@5 | MRR |  Recall@5 |
> | 1                   |  25.87              | 37.43              |39.33           | 44.14           |
> | 2                   |  28.08              | 39.25              |55.24           | 65.32           |
> | 3                   |  28.39              | 39.57              |56.50           | 66.79           |
> | **4**                   |  28.14              | 38.37              |57.41           | 68.22           |
> | **5**                   |  28.70              | 39.26              |57.86           | 68.79           |
>
> *Table R9*: Effect of **iterative** optimization in DualReform on retrieval accuracy and pseudo reference accuracy. Please refer to the [`🔗link`](https://drive.google.com/file/d/14-rr6374MkmyaAlxa3apOzkrii7gasJO/view?usp=sharing) for a visual overview of the trends.
>
>
>
>
> ---
>
> `W1. Could the authors provide easy but intuitive brief introduction of Lemmas 4.2 and 4.3? I know that most of them may have been well validated in prior works, but the reader of paper may not familiar with them.`
>
> We thank the reviewer for pointing out that Lemmas 4.2 and 4.3 would benefit from a more intuitive introduction, and we apologize for any confusion caused by the current presentation. Conceptually, DualReform implements a self-training cycle [Amini et al., 2025], in which generating pseudo reference passages and training the CQR model reinforce each other in an iterative loop. Within this framework, Lemma 4.2 shows that after a **single** round of training the CQR model on pseudo reference passages, the generalization bound can already be improved. Lemma 4.3 then extends this result to the **iterative** setting: when we repeatedly alternate pseudo reference passage generation and CQR training, as in the DualReform, the generalization bound is further tightened.
>
> Intuitively, Lemma 4.2 (adapted from [Wei et al., 2021]) formalizes the standard “student–teacher” view of self-training: a **student** model trained on pseudo labels produced by a teacher **surpasses** the teacher. More precisely, if Assumption 4.1—akin to the widely used smoothness assumption [Chapelle et al., 2006]—holds, then optimizing toward pseudo labels with an input-consistency loss forces the model into correcting the erroneous labels and thereby enhance training accuracy. In DualReform, this input-consistency loss is enforced through various input transformations, such as multiple query reformulations generated by an LLM, which encourage stable predictions across diverse input forms.

---

> > ### Author Response · Authors · 2025-11-21
> >
> > `W2. I do not directly focus on this research field, so I have a concern about the definition of reference-free. From my own perspective, this work is still reference-based because, in step 2, this work still needs the reference, although this reference is a pseudo-reference. Thus, compared to traditional reference-based methods, the major difference is an automatic way to label the reference.`
> >
> > We sincerely appreciate the reviewer’s concern and understand why the term “reference-free” can be confusing. From the reviewer’s perspective, it is indeed natural to regard our method as still reference-based, since the model is trained using **pseudo reference passages** rather than operating without any form of reference at all.
> >
> > In this work, our use of “reference-free” is intended to emphasize that DualReform does **not rely on *human-annotated* reference passages**. DualReform assumes a a reference-free (label-free) conversational setting and instead automatically generates **pseudo reference passages** using the given retriever and CQR model. This is analogous to **self-supervised learning** relying on **pseudo-labeling** methods, where one still trains on “labels”, but these labels are produced automatically from the data and the model rather than provided as external supervision; in this sense, such methods are often described as **“label-free”** optimization [Ferreira et al., 2025]. We will clarify this terminology in the discussion (e.g., with “human-reference-free / pseudo-reference-based training) and with a clear analogy to self-supervised learning and pseudo-labeling, so that readers who are less familiar with this line of work are not misled by the terminology.
> >
> >
> > ---
> >
> > `W3. In Table 6 (Sec 5.4), it would be better to use more reference-free metrics (just like LLMEval) to evaluate the generated dialogues along with the current reference-aware metrics.`
> >
> > We thank the reviewer for this suggestion and agree that combining reference-based and reference-free metrics yields a more complete assessment of dialogue quality. In response, we additionally report widely used **reference-free LLMEval metrics** [Zheng et al., 2023], which evaluate generated responses without ground-truth references. Specifically, Table R10 includes **(a) LLMEval–Helpfulness** and **(b) LLMEval–Correctness**; given that our problem setting is conversational search / information retrieval, we focus on these criteria as they directly capture *usefulness* and *factual accuracy* of answers.
> >
> > Overall, DualReform outperforms all CQR baselines not only on the reference-based metrics in the manuscript, but also on the new reference-free metrics. These results further support that DualReform improves dialogue quality by enabling more accurate passage retrieval. We will integrate these results and the corresponding discussion into Table 6 and Sec. 5.4 in the revised manuscript.
> >
> >
> > |  |  **Reference-free Metrics (new)** |       |        Reference-Based Metrics (already in Table 6)             |   |
> > |-------------|--------------------------------|------------------------------|--|--------------|
> > |        CQR Methods    | **(a) LLMEval–Helpfulness** (↑) | **(b) LLMEval–Correctness** (↑) | LLMEval-Reference (↑) | ROUGE-1 (↑) |
> > | LLM-IQR     | 46.82                     | 58.19                      | 27.76             | 20.07       |
> > | HyDE-LLM    | 54.85                     | 66.89                      | 29.77             | 22.18       |
> > | LLM4CS-CoT  | 59.20                     | 67.22                      | 26.42             | 20.23       |
> > | T5QR        | 44.82                     | 60.87                      | 28.43             | 19.96       |
> > | ConvGQR     | 51.51                     | 62.88                      | 23.08             | 20.81       |
> > | HyDE-FT     | 49.81                     | 66.80                      | 23.17             | 19.81       |
> > | RetPO       | 52.17                     | 63.21                      | 27.09             | 21.63       |
> > | **DualReform (ours)** | **62.54**               |**70.57**         |  **34.45** | **24.37** |
> >
> > *Table R10*: Response generation quality with passages retrieved by different CQR methods, evaluated using reference-based and **reference-free** metrics (LLMEval-helpfulness, LLMEval-correctness).
> >
> > ---
> > ---
> > [Xie et al., 2020] Self-Training with Noisy Student Improves ImageNet Classification. CVPR, 2020
> >
> > [Chapelle et al., 2006] Semi-Supervised Learning. MIT Press, 2006
> >
> > [Wei et al., 2021] Theoretical Analysis of Self-Training with Deep Networks on Unlabeled Data. ICLR, 2021
> >
> > [Amini et al., 2025] Self-Training: A Survey. Neurocomputing, 2025
> >
> > [Ferreira et al., 2025] Self-supervised learning for label-free segmentation in cardiac ultrasound. Nature, 2025
> >
> > [Zheng et al., 2023] Judging LLM-as-a-Judge with MT-Bench and Chatbot Arena. NeurIPS, 2023

---

> > > ### Author Response · Authors · 2025-11-25
> > > **Kindly Follow-Up**
> > >
> > > Dear Reviewer Kn6Q:
> > >
> > > We greatly appreciate your thoughtful feedback and the time devoted to this review, which substantially improved the quality of our work. If possible, we would be grateful for any additional insights at your convenience, as they would help us further improve this work. The authors would also be happy to discuss any aspects in more detail. Thank you very much.
> > >
> > > Sincerely,
> > >
> > > Authors

---

> > > > ### Author Response · Authors · 2025-12-01
> > > > **For Area Chair: Summary of Responses to Reviewer Kn6Q**
> > > >
> > > > The reviewer raised four requests: (i) the performance trend beyond 3 iterations, (ii) a more intuitive presentation of Lemmas 4.2–4.3, (iii) an unambiguous definition of "reference-free," and (iv) reference-free response evaluation in addition to the existing metrics. In the rebuttal, we provide additional evidence and clarifications that resolve the concerns by extending the iterations, strengthening the explanation of the lemmas, disambiguating "reference-free," and adding reference-free response-quality metrics (Tables R9–R10).
> > > >
> > > >
> > > > | Comment | Concern                  | Response                                                                                                                            |
> > > > | ------- | ------------------------ | ----------------------------------------------------------------------------------------------------------------------------------- |
> > > > | Q1      | Additional model analysis       | **Additional results.** We extend DualReform to **5 iterations** and reconfirm the saturation beyond 3 iterations. See **Table R9**.                |
> > > > | W1      | Intuition of Lemma 4.2          | **Clarification.** We add an intuitive explanation of Lemmas 4.2–4.3 via the self-training (student–teacher) view.                     |
> > > > | W2      | Terminology clarity | **Clarification.** We clarify "reference-free" as **human**-reference-free, and motivate the term via an analogy to self-supervised learning, often called "label-free," that uses pseudo-labels.               |
> > > > | W3      | Evaluation coverage   | **Additional results.** We add **reference-free** LLMEval response metrics and report the results, reconfirming DualReform's effectiveness. See **Table R10**. |
> > > >
> > > > We believe this rebuttal addresses the reviewer's concerns, and the reviewer would therefore be inclined to increase his/her rating.

---

### Official Review · Reviewer_YCPm · 2025-11-01

**Soundness:** 3
**Presentation:** 3
**Contribution:** 3
**Rating:** 6
**Confidence:** 4

**Summary:**

This paper proposes DualReform, a reference-free preference-optimization pipeline for conversational query reformulation (CQR). It infers pseudo reference passages by refining responses with the CQR model, then uses those pseudo references to generate preference feedback and run SFT+DPO, showing strong retrieval and generation gains close to a reference-based upper bound.

**Strengths:**

1. Empirical gains are substantial and consistent across datasets and retrievers, often nearing the upper-bound.
2. The iterative refinement and query-forming template are well studied with ablations and extended metrics, showing clear contributions from each design choice.

**Weaknesses:**

1. Component-level attribution could be better: the paper reports useful ablations but does not present a unified per-module breakdown (retriever type, pseudo refs, etc.,) to show which component drives most of the gains.
2. Iterative pseudo-refinement and extra retrieval steps likely increase per-query compute. Practical cost and latency are necessary but unreported.
3. The training pipeline relies on reformulated queries generated by ChatGPT. Can it be achieved using reformulations produced by the trained model itself instead of an external LLM？

**Questions:**

See weakness 3.

---

> ### Author Response · Authors · 2025-11-21
>
> We sincerely appreciate the reviewer’s valuable time and constructive feedback throughout the review process. We hope the clarifications in this rebuttal sufficiently address any remaining questions.
>
> `Q1/W3. The CQR training pipeline relies on reformulated queries generated by ChatGPT. Can we achieve CQR training using query reformulations produced by the trained model itself？`
>
>
> We completely agree with the reviewer’s question and in fact had the same question ourselves, which is why we mentioned this point in the Limitation section. Building on the reviewer’s suggestion, we implement a **new** variant, **DualReform+1R-Self**, in which the additional CQR training stage uses query reformulations **generated by the trained CQR model itself**. Concretely, we first take the CQR model trained by DualReform, use it to generate additional candidate query reformulations, and then continue training the model on these self-generated candidates. For a fair comparison, we also introduce **DualReform+1R-GPT** variant (using ChatGPT instead of the trained CQR model), which uses the same additional training budget but relies on the ChatGPT-generated reformulations.
>
> Overall, Table R6 shows that DualReform+1R-Self consistently achieves **higher** performance than DualReform+1R-GPT, achieving relative gains of up to 4.16%. This indicates that CQR training can indeed be successfully driven by query reformulations produced by the trained model itself. In summary, as the reviewer anticipated, this approach is not only effective but also cost-effective, since it **avoids additional external LLM calls** once the CQR model has been trained.
>
> |                   | Sparse     |   |   |   | Dense     | |||
> |----------------------------------------|---------------|--------------|--------------|--------------|--------------|-------------|-------------|-------------|
> | CQR Methods                  | MRR    | NDCG  | R@5   | R@20  |  MRR    |  NDCG  |  R@5   |  R@20  |
> | Upper Bound                 | 29.19±0.30    | 27.19±0.22   | 39.72±0.87   | 56.89±0.97   | 40.58±0.22   | 39.15±0.25  | 53.42±1.04  | 72.69±1.22  |
> | RetPo               | 23.20±0.33    | 21.41±0.28   | 32.18±0.20   | 48.54±0.22   | 35.67±0.05   | 34.28±0.04  | 49.99±0.31  | 67.47±0.10  |
> | DualReform                            | 28.39±0.39    | 26.08±0.46   | 39.57±1.05   | 56.62±1.24   | 40.14±0.21   | 38.33±0.28  | 53.78±0.58  | 71.99±0.30  |
> | DualReform + **1R-GPT**           | 28.14±0.47    | 26.14±0.51   | 38.36±0.44   | 55.95±0.31   | 40.72±0.17    | 39.38±0.18   | 54.15±0.45   | 72.98±0.83 |
> | DualReform + **1R-Self**           | **29.22±0.17**    | **26.93±0.33**   | **39.25±1.02**   | **58.28±0.41**   | **42.27±0.57**   | **40.79±0.65**  | **56.21±0.46**  | **74.53±0.17**  |
>
> *Table R6*: Retrieval performance comparison, where **DualReform+*1R-Self*** additionally incorporates reformulated queries produced by the trained CQR model itself as candidate reformulations.

---

> ### Author Response · Authors · 2025-11-21
>
> `W1. Could the authors provide component-level attribution? The paper reports useful ablations but does not present a unified per-module breakdown to show which component drives most of the gains.`
>
> We thank the reviewer for acknowledging the usefulness of our ablations and are pleased to provide a unified attribution. Table R7 provides a per-component breakdown of DualReform for generating pseudo reference passages by progressively activating (●): (i) **retrieval of pseudo reference passages** that enables our reference-free approach, (ii) **LLM-based response refinement**, and (iii) **CQR-based response refinement**. Overall, activating pseudo reference retrieval (Variant 1 → Variant 2) yields the largest performance increase, indicating that the reference-free approach itself is the primary driver of the gains. The subsequent response refinement stages, particularly the transition from Variant 3 to DualReform (activating CQR refinement), provide further noticeable gains by improving the quality of pseudo reference inference via higher-quality responses.
>
> |  | Main Components |  | | TopiOCQA | | SciConvQA | |
> |--------:|:---------:|:----------:|:---------:|---------:|-----:|----:|----:|
> | Variant | **CQR Response Refinement** | **LLM Response Refinement** | **Pseudo Reference Retrieval** | Recall@5 | MRR | Recall@5 | MRR |
> | **Variant 1**  | ○          | ○          | ○             | 32.18        | 23.20         | 21.61    |  15.60    |
> | **Variant 2**  |○          | ○          | ●            | 37.43        | 25.87        |26.05    |  17.55    |
> | **Variant 3**  | ○          | ●          | ●              | 37.79        | 27.92        |26.77    |  19.02    |
> | **DualReform** | ●      | ●          | ●         | **39.57**    | **28.39**    | **28.64**    |  **20.06**    |
>
> *Table R7*: Component-level ablation of DualReform. Retrieval performance on TopiOCQA and SciConvQA is reported as we progressively enable (●) pseudo reference retrieval, LLM response refinement, and CQR response refinement (Variants 1–3), with DualReform using all three components.
>
>
> ---
>
>
> `W2. Iterative pseudo-refinement and extra retrieval steps likely increase per-query compute. What is practical cost and latency?`
>
> We appreciate the reviewer’s attention to latency and computational cost. DualReform introduces additional computation **only in the offline training pipeline**, **in exchange for removing the need for costly manual annotation of reference passages**. Once the CQR model is trained, **inference-time** latency is effectively **identical** to that of the underlying CQR baseline, since DualReform does not invoke any extra LLM calls or additional retrieval steps at test time.
>
> Compared to reference-based methods such as RetPO [Yoon et al.,2024], the extra cost in the training stage comes from three components: (i) response refinement via CQR (Eq. (9)), (ii) a retrieval step to obtain pseudo reference passages (Eq. (8)), and (iii) DPO training on the resulting pseudo reference passages (Eq. (10)). As summarized in Table R8, the overall overhead required by these steps is **moderate** and largely dictated by the available GPU resources. In particular, using a **single NVIDIA RTX A6000** requires only an additional **6.41 hours**, with the main overhead stemming from stages (i) and (iii).
>
>
> |           | (i) Response Refinement via CQR | | (ii) Retrieval of Pseudo Reference Passages | | (iii) DPO Training | |
> |------------------|--------------|-------------|-------------|--------------|----------|--|
> |      | A6000  | A100 (est.) | Sparse Retrieval  | Dense Retrieval | A6000  | A100 (est.)    |
> | Wall-clock time | 3.58h (1.289s/query)|1.77h (0.639s/query)| 1.15mins (0.015s/query)| 2.42mins (0.007s/query)| 2.79h | 1.43h|
>
> *Table R8*: Wall-clock time for each major component of DualReform on SciConvQA. Time is reported in hours (h), minutes (min), and seconds (s) for a single RTX A6000 GPU. The "A100 (est.)" columns denote approximate wall-clock times on an NVIDIA A100 GPU, obtained by scaling the A6000 measurements using a 2.02× speedup factor following prior work [Massed Compute, 2024].
>
>
> ---
>
>
> [Yoon et al.,2024] Ask Optimal Questions: Aligning Large Language Models with Retriever’s Preference in Conversational Search. arXiv, 2024
>
>
> [Massed Compute, 2024] LLama 3 Benchmark Across Various GPU Types. Massed Compute. 2024. Available at: https://massedcompute.com/llama-3-benchmark-across-various-gpu-types/

---

> > ### Author Response · Authors · 2025-11-25
> > **Kindly Follow-Up**
> >
> > Dear Reviewer YCPm:
> >
> > We greatly appreciate your thoughtful feedback and the time devoted to this review, which substantially improved the quality of our work. If possible, we would be grateful for any additional insights at your convenience, as they would help us further improve this work. The authors would also be happy to discuss any aspects in more detail. Thank you very much.
> >
> > Sincerely,
> >
> > Authors

---

> > > ### Author Response · Authors · 2025-12-01
> > > **For Area Chair: Summary of Responses to Reviewer YCPm**
> > >
> > > The reviewer's main requests center on (i) reducing reliance on external LLM-generated reformulations, (ii) clearer component-level attribution, and (iii) practical compute/latency reporting. In the rebuttal, we directly resolve these concerns with new evidence, showing that self-generated reformulations can effectively mitigate ChatGPT reliance, providing clearer module-level attribution, and showing that the overall overhead is very small (Tables R6–R8).
> > >
> > >
> > >
> > > | Comment | Concern | Response |
> > > |---|---|---|
> > > | W3, Q1 | ChatGPT reliance | **Additional results.** We compare two training variants based on **self-generated vs. ChatGPT-generated** reformulations; the self-generated variant performs better. See **Table R6**.|
> > > | W1 | Component-level attribution | **Additional results.** We provide a unified per-module breakdown, showing that the main gains come from our reference-free component. See **Table R7**. |
> > > | W2 | Cost/latency analysis | **Clarification+Additional results.** Extra compute is offline-only (no inference-time impact), and the measured training overhead is **very small**. See **Table R8**. |
> > >
> > > We believe this rebuttal addresses the reviewer's concerns, and the reviewer would therefore be inclined to increase his/her rating.

---

### Official Review · Reviewer_UcSN · 2025-11-01

**Soundness:** 2
**Presentation:** 2
**Contribution:** 2
**Rating:** 2
**Confidence:** 5

**Summary:**

The paper presents a framework for reference-free preference optimization in CQR. It generates pseudo reference passages directly from conversational datasets instead of relying on manually curated reference passages. By using the refined responses for retrieval, more accurate pseudo passages can be obtained. These pseudo passages are used to construct preference feedback. Additionally, the paper proposes an iterative optimization scheme that alternates between pseudo passage retrieval and preference optimization, enabling the pseudo passages and CQR model to continuously enhance.

**Strengths:**

1. By iterating between pseudo passages retrieval and preference optimization through a self-reinforcement loop, both can promote each other, leading to continuous optimization of the CQR model.
2. Similar to query reformulation, the response is refined to clarify ambiguities and omissions, leading to more accurate retrieval of pseudo passages.
3. Experiments demonstrate that DUALREFORM achieves retrieval accuracy very close to that of systems using true references.

**Weaknesses:**

1.  The motivation and novelty of this research is not convincing. There is no necessity for inferring pseudo passages just for training, especially limited to the conversational QR scenarios. The community already has many datasets with relevance judgments for such a purpose. Besides, the definition of reference is unclear here. If this denotes relevance judgments, then previous CQR methods do not necessarily use relevance judgments, which means the claim in the second paragraph in the Introduction is not correct. If this does not refer to relevance judgments, e.g., the response or sth else, then what is the scenario in practice requires it? In addition, preference optimization without reference should be able to completely eliminate the dependence on passage collection during the training phase.

2. The proposed model is both trained and evaluated on in-domain conversations, whereas other CQR baselines are trained on out-of-domain data and evaluated on the in-domain dataset. Although our model does not rely on annotated relevant passages during training, its alignment with the evaluation data distribution may partially account for its performance advantages over models trained under distributional mismatch. Besides, the comparison in the experiment is unfair. For ConvGQR, HyDE-FT, and RetPO, they do not have access to the target dataset's passage collection during training, while DUALREFORM uses the target dataset's passage collection (i.e., the pseudo passages retrieved) during training. Another unfair comparison in the experiment is the use of GTR-Large in dense retrieval, while most previous methods used ANCE. The performance improvement in dense retrieval may come from the retriever.

3. Lack of manual evaluation to compare the difference in quality between pseudo passages and ground-truth ones and some baselines are missing, e.g., CHIQ [1] and AdaCQR [2].

[1] CHIQ: Contextual History Enhancement for Improving Query Rewriting in Conversational Search

[2] AdaCQR: Enhancing Query Reformulation for Conversational Search via Sparse and Dense Retrieval Alignment

**Questions:**

1. What are the differences compared to CHIQ-FT in CHIQ: Contextual History Enhancement for Improving Query Rewriting in Conversational Search, which also tries to improve the quality of relevance judgments in terms of CQR training to improve the performance of conversational search.

2. What is the latency analysis, for both data generation and inference? The method relies heavily on LLMs.

3. Similar to weakness one, what is the motivation to generate new reference but discarding the relveance judgments in the training set? If this is toward a general goal, why limited only in conversational query reformulation but not to general zero-shot dense retrieval, e.g., evaluating on BEIR? I don't see any strong connection with the research question in conversational QR.

---

> ### Author Response · Authors · 2025-11-19
>
> We sincerely appreciate the reviewer’s valuable time and feedback throughout the review process. We believe there are a few misunderstandings, and hope that our clarifications in this rebuttal will thoroughly address them.
>
> `W1. There is no necessity for inferring pseudo passages just for training, especially limited to the conversational QR scenarios. The community already has many datasets with relevance judgments for such a purpose. Besides, the definition of reference is unclear here. If this denotes relevance judgments, then previous CQR methods do not necessarily use relevance judgments, which means the claim in the second paragraph in the Introduction is not correct. If this does not refer to relevance judgments, e.g., the response or sth else, then what is the scenario in practice requires it?`
>
>
> ***What are "reference passages" and "relevance judgments"?***
>
> Reference passages and relevance judgments play fundamentally **different** roles. **Reference passages** termed **gold/ground-truth passages** are used to evaluate retrieval quality as the ideal retrieval targets. **Relevance judgments,** termed “Pseudo Relevance Labels (PRJs)” [Mo et al., 2023; 2024a, b] provide a **supervision signal**  that guides the CQR model to select the *informative context*. Importantly, constructing these relevance judgments **requires** reference passages to assess how much each historical turn contributes to retrieval quality:
> - **How are relevance judgments computed?** Following the notations of [Mo et al., 2024a], given a current query $u_n$ and its associated reference passage $p_n^\*$, relevance judgments are the relevance of historical turns $\\{(u\_i, p\_i^\*)\\}\_{i=1}^{n-1}$. A historical turn $(u_i, p_i^\*)\_{i<n}$ is judged relevant if adding $(u_i, p_i^\*)_{i<n}$ leads to an “improved retrieval performance” for $u_n$.
> - **Where are reference (gold) passages used?** Evaluating “improved retrieval performance” requires measuring how accurately **reference passage $p_n^\*$** is retrieved from the database. Concretely, the improvement in retrieval score is computed as $score((u_i, p_i^\*, u_n),p_n^\*)-score(u_n,p_n^\*)$, and a historical turn $(u_i, p_i^\*)$ is deemed relevant if this difference is positive. Therefore, reference passages are the fundamental resource required to produce relevance judgments.
>
>
> ```
> **Example**: How are relevance judgments⚖️ computed using reference passages📌?
>
> Conversation History          ─▶ Compute Relevance Judegements ⚖️ with Reference📌
> ─────────────────────────────────────────────────────────────────────────────────────────────────────────────────
> ┌────────────────────────────┐  ┌───────────────────────────────────────────────┐
> │ Turn 1                     │  │ Add (u₁, p₁*) to current query u₃             │
> │ u₁: "Who is Marie Curie?"  │─▶│→ retrieval of p₃*📌 improves                  │
> │ p₁*📌: [biography of Curie]│  │→ Relevance Judegements-Turn 1⚖️: ✅relevant   │
> │ └ ⚠️ Mostly Unavailable    │  └───────────────────────────────────────────────┘
> └────────────────────────────┘
> ┌────────────────────────────┐  ┌───────────────────────────────────────────────┐
> │ Turn 2                     │  │ Add (u₂, p₂*) to current query u₃             │
> │ u₂: "Where was she born?"  │─▶│→ no improvement in retrieving p₃*📌           │
> │ p₂*📌: [place of birth]    │  │→ Relevance Judegements-Turn 2⚖️: ❌irrelevant │
> │ └ ⚠️ Mostly Unavailable    │  └───────────────────────────────────────────────┘
> └────────────────────────────┘
> Current Query and Corresponding Reference (Gold/GT) Passage📌
> ┌───────────────────────────────────────────────────────────────────────────────┐
> │ Turn 3 (current)                                                              │
> │ u₃: "When was her second Nobel Prize awarded?"                                │
> │ p₃*📌: [passage about second Nobel Prize]  ← reference passage used for u₃    │
> │ └ ⚠️ Mostly Unavailable                                                       │
> └───────────────────────────────────────────────────────────────────────────────┘
> ```

---

> > ### Author Response · Authors · 2025-11-19
> >
> > (***Continued***) `W1. There is no necessity for inferring pseudo passages just for training, especially limited to the conversational QR scenarios. The community already has many datasets with relevance judgments for such a purpose. Besides, the definition of reference is unclear here. If this denotes relevance judgments, then previous CQR methods do not necessarily use relevance judgments, which means the claim in the second paragraph in the Introduction is not correct. If this does not refer to relevance judgments, e.g., the response or sth else, then what is the scenario in practice requires it?`
> >
> > ***What practical scenario requires inferring pseudo reference passages? Why do we need them?***
> >
> > Reference passages are the fundamental resource for making relevance judgments. Yet **recent large-scale conversational datasets** [Zheng et al., 2023; Ding et al., 2023; Zhao et al., 2024] do **not** include annotated reference passages. As summarized in Table R1, these datasets (e.g., WildChat, UltraChat, LMSYS-Chat, ShareGPT, OASST1) already amount to **7.3 million** conversations in total. While the **main conversational search benchmarks** (e.g., QReCC, TopiOCQA) include such reference passages, they cover only **17K** conversations, i.e., more than 400 times fewer, and are confined to popular Wikipedia topics [Anantha et al., 2021]. It is therefore unsurprising that models trained only on these limited resources **generalize poorly** to real user scenarios with diverse domains, intents, and conversational styles.
> >
> > To develop CQR models that generalize to such real-world usage, we ultimately **need reference passages** for the kinds of conversations that actually occur in practice, i.e., within these **large-scale** conversational datasets. However, manually annotating reference passages for millions of queries is prohibitively expensive, requiring substantial human effort and domain expertise. This motivates our work: we propose a cost-efficient and effective approach to infer pseudo reference passages from recent large-scale conversational datasets, thereby bridging the gap between abundant real-world conversations and the scarce supply of annotated reference passages.
> >
> > | Dataset              | **#Conversations (K)** | **Reference passages** | Characteristics                                              |
> > |----------------------|------------------------|------------------------|------------------------------------------------------------------|
> > | **Recent large-scale conversational datasets** ||||
> > | WildChat-4.8M-Full   | 4,800                  | ✗                      | In-the-wild ChatGPT logs across diverse domains                  |
> > | UltraChat            | 1,400                  | ✗                      | Multi-turn dialogues mimicking user scenarios          |
> > | LMSYS-Chat-1M        | 1,000                  | ✗                      | Chatbot Arena logs over multiple LLMs                       |
> > | ShareGPT             | 90                     | ✗                      | User-shared ChatGPT conversations                                |
> > | OASST1               | 66                     | ✗                      | Crowd-sourced dialogues with human ratings       |
> > | *[All-Future-Datasets]*| ∞                      | ✗ (rare)               | *[Continuously growing with new domains and emerging topics]*  |
> > | **Conversational search benchmarks** ||||
> > | QReCC                | 13.7                   | ✓                      | Combined from QuAC, TREC CAsT, and NQ             |
> > | TopiOCQA             | 3.5                    | ✓                      | Topics grounded in Wikipedia                          |
> >
> > *Table R1*. Recent large-scale conversational datasets versus conversational search benchmarks, showing the number of conversations and whether gold reference passages are annotated (K = thousands of conversations; ✓/✗ indicate the presence/absence of annotated reference passages).

---

> > > ### Author Response · Authors · 2025-11-19
> > >
> > > `Q1. What are the differences compared to CHIQ-FT in CHIQ: Contextual History Enhancement for Improving Query Rewriting in Conversational Search, which also tries to improve the quality of relevance judgments in terms of CQR training to improve the performance of conversational search.`
> > >
> > > We appreciate your suggestion to compare our work with the great work CHIQ-FT [Mo et al., 2024a]. While both approaches are concerned with using relevance judgements in CQR training, they operate under **different assumptions about reference passages**.
> > >
> > > CHIQ-FT explicitly assumes **access** to reference (gold) passages and, as shown in Equations (2) and (3) of [Mo et al., 2024a], leverages these reference passages to construct accurate relevance judgments for CQR training. Similarly, preference-optimization–based CQR methods such as RetPO [Yoon et al., 2024] rely on reference passages to derive preferences that are another form of relevance judgments. By contrast, our work focuses on the more practical setting where reference passages are **not available**. To enable the  construction of relevance judgements in this *reference-free* regime, we propose an approach that generates **pseudo** reference passages directly from conversational data and then uses them to derive relevance judgments. We will revise the manuscript to include this comparision with CHIQ-FT.
> > >
> > >
> > > ---
> > > `Q3. What is the motivation to generate new reference but discarding the relevance judgments in the training set?`
> > >
> > > We would like to clarify that our goal is **not** to discard useful relevance judgments. On the contrary, we introduce pseudo reference passages precisely to **obtain** relevance judgments in settings where reference passages are absent. Specifically, we first infer **pseudo reference passages** from conversational data, and then use them to obtain preference signal, which is another form of relevance judgments.
> > >
> > >
> > > ---
> > >
> > > `W2-1. The comparison in the experiment is unfair. The proposed model is both trained and evaluated on in-domain conversations and passage collection, whereas other CQR baselines are trained on out-of-domain data and evaluated on the in-domain dataset.`
> > >
> > > We deliberately design our experiments to reflect a **realistic setting where reference passages are not available** on the target dataset, as handling this constraint is precisely the practical scenario our method is intended to address. Thus, for baselines that **require** reference passages (e.g., RetPO), training on the target dataset (e.g., TopiOCQA) is infeasible because reference passages are not provided. We therefore follow a practical setup: these methods are trained on QReCC, whose domains and passage collection are **similar** to those of TopiOCQA (see Figure 9), and evaluated on TopiOCQA. In contrast, for methods that do **not require** reference passages (e.g., T5QR) and for our method, we **do** train on the target dataset.
> > >
> > > Furthermore, we even evaluate **our work under less favorable** conditions by including the **Upper Bound** baseline that has **access to *genuine reference passages*** on the target dataset and is *trained* and *evaluated* on in-domain conversations and the target passage collection. This configuration represents an *ideal but unrealistic* scenario with full reference passages, and thus deviates from the realistic setting we target. Nevertheless, our work achieves performance very close to this Upper Bound while *not* using any reference passages on the target dataset.

---

> > > > ### Author Response · Authors · 2025-11-19
> > > >
> > > > `W2-2. Unfair comparison in the experiment is the use of GTR-Large in dense retrieval, while most previous methods used ANCE. The performance improvement in dense retrieval may come from the retriever. `
> > > >
> > > >
> > > > We would like to reassure the reviewer that this concern does **not** arise in our experimental setup. In our experiments, GTR-Large [Ni et al., 2022] is used as the dense retriever for **all** methods, including the baselines and DualReform, meaning no method is provided with a stronger retriever than the others.
> > > >
> > > > Although ANCE was not employed in the our results, we **additionally** conducted the dense-retrieval experiments using **ANCE** [Xiong et al., 2020] for every method to directly address the reviewer’s concern. As shown in Table R2, ANCE achieves roughly 8–10% **lower** average performance on SciConvQA compared to GTR-Large. We suspect that this is because GTR-Large, being a more recent model, is more extensively trained for the specialized scientific domain of SciConvQA. Conversely, on the more general TopiOCQA dataset, GTR-Large and ANCE perform **similarly**, suggesting that both retrievers are sufficiently trained for broad, non-specialized topics.
> > > >
> > > >
> > > > | Dataset | CQR  | MRR |  | nDCG |  | R@5 | |
> > > > |---------|------|-----|-----|------|------|-----|-----|
> > > > |         |      | GTR | **ANCE** | GTR  | **ANCE** | GTR | **ANCE** |
> > > > | SciConvQA| HyDE-LLM          | 16.70±0.69  | 14.99±0.17     | 15.38±0.66  | 14.04±0.33      | 23.28±0.70  | 20.93±0.34     |
> > > > |          | LLM4CS-CoT        | 18.25±0.22  | 16.68±0.10     | 16.75±0.18  | 15.56±0.10      | 24.48±0.22  | 22.33±0.37     |
> > > > |          | RetPO             | 17.95±0.08  | 16.30±0.25     | 16.74±0.08  | 15.29±0.21      | 24.31±0.36  | 21.46±0.25     |
> > > > |          | DualReform (ours) | 23.53±0.09  | 20.29±0.18     | 22.04±0.23  | 19.67±0.09      | 31.19±0.18  | 27.93±0.29     |
> > > > | -        | **average degradation (%)** |   -  | **-10.45**             |  -    |**-8.81**          |    -  | **-10.26**            |
> > > > | TopiOCQA | HyDE-LLM          | 33.39±1.48  | 31.83±1.11     | 32.02±1.34  | 30.54±1.39      | 45.54±0.05  | 42.39±0.65     |
> > > > |          | LLM4CS-CoT        | 37.55±0.75  | 38.91±1.04     | 35.92±1.04  | 38.18±1.03      | 52.45±0.35  | 51.95±0.61     |
> > > > |          | RetPO             | 35.67±0.05  | 35.29±0.23     | 34.28±0.04  | 34.35±0.30      | 49.99±0.31  | 48.55±0.40     |
> > > > |          | DualReform (ours) | 40.14±0.21  | 39.11±0.60     | 38.33±0.28  | 38.38±0.72      | 53.78±0.58  | 51.90±0.84     |
> > > > | -        | **average degradation (%)** |  -   | -1.17              |    -  | +0.50              |   -   | -3.56              |-
> > > >
> > > >
> > > > *Table R2*: Comparison of two **dense retrievers** **(GTR vs. ANCE)**, with the last row in each block showing the **average degradation of ANCE w.r.t. GTR**, computed as (ANCE−GTR)/GTR×100%.

---

> ### Author Response · Authors · 2025-11-19
>
> `W3. Lack of manual evaluation to compare the difference in quality between pseudo passages and ground-truth ones and some baselines are missing, e.g., CHIQ [Mo et al.,2024a] and AdaCQR [Lai et al., 2025].`
>
> We agree that evaluating the quality of pseudo reference passages is crucial and have addressed it through both quantitative and qualitative analyses. Quantitatively, we measure **pseudo reference accuracy**, which captures the agreement between our pseudo reference passages and the **ground-truth reference passages**. Table R3 compares pseudo reference accuracy of DualReform’s pseudo reference passages with those generated by alternative LLMs. DualReform attains higher pseudo reference accuracy, indicating that our generated references closely match the ground-truth documents. The complete results appear in Table 3 of the manuscript.
>
> | Refine LLMs | MRR | NDCG | Recall@5 | Recall@20 |
> | -------------- | --------------- | --------------- | ------------------ | ------------------ |
> | Llama          | 36.55           | 35.62           | 46.13              | 59.37              |
> | Llama+ICL      | 44.82           | 43.99           | 54.13              | 66.00              |
> | **DualReform** | **50.05**       | **49.36**       | **59.75**          | **71.27**          |
>
> *Table R3*: Pseudo reference accuracy of pseudo reference passages generated by DualReform and alternative LLMs. All variants, including DualReform, use Llama-3-8B-Instruct as the backbone model and share the same prompt (Prompt 1 in the manuscript). The Llama+ICL variant additionally applies in-context learning.
>
> We also conduct a qualitative analysis: Figures 5 and 15–17 in the manuscript present concrete examples where the **responses refined by DualReform** closely match the **ground-truth reference passages**. Recall that DualReform refines each response in a context-aware manner; the refined response is then used to retrieve pseudo reference passages (e.g., via BM25), as detailed in Eqs. (8) and (9). Overall, the examples confirm that the refined responses capture the essential semantics of the ground-truth passages.
>
>
> In response to the reviewer’s constructive suggestion regarding additional baseline, we further include the recent baseline, **AdaCQR** [Lai et al., 2025] in our experiments. Table R4 reports a comparison between DualReform and CQR baselines, now including AdaCQR. Since AdaCQR, as with RetPO, is designed to *leverage reference passages* for producing preference signals, its performance without reference passages is similar to RetPO and remains clearly below DualReform on the target dataset. This underscore the benefit of our reference-free approach in scenarios where reference passages are not available.
>
> | Data      | Query Reformulations                  | Sparse MRR    | Sparse NDCG  |  Dense MRR    | Dense NDCG  |
> |-----------|----------------------------------------|---------------|--------------|--------------|--------------|
> | SciConvQA | Upper Bound                   | 20.89±0.30    | 18.91±0.58   |  23.73±0.70   | 22.49±0.80  |
> |  | LLM4CS-CoT                             | 16.53±0.18    | 15.29±0.27   |  18.25±0.22   | 16.75±0.18  |
> |  | RetPo                                  | 15.60±0.23    | 14.36±0.24   |  17.95±0.08   | 16.74±0.08  |
> |  | **AdaCQR**                                  | 15.97±0.31    | 13.97±0.34   |  20.14±0.17   | 18.83±0.19  |
> |  | **DualReform** (ours)                      | 20.06±0.19    | 18.42±0.27   |  23.53±0.09   | 22.04±0.23  |
> | TopiOCQA  | Upper Bound                   | 29.19±0.30    | 27.19±0.22   |  40.58±0.22   | 39.15±0.25  |
> |   | LLM4CS-CoT                            | 26.81±0.66    | 25.40±0.38   |  37.55±0.75   | 35.92±1.04  |
> |   | RetPo                                 | 23.20±0.33    | 21.41±0.28   |  35.67±0.05   | 34.28±0.04  |
> |   | **AdaCQR**                                 | 24.78±0.63    | 22.60±0.78   |  36.45±0.50   | 35.20±0.56  |
> |   | **DualReform** (ours)                     | 28.39±0.39    | 26.08±0.46   |  40.14±0.21   | 38.33±0.28  |
>
> *Table R4*: Retrieval accuracy on SciConvQA and TopiOCQA with representative CQR baselines, extended to include the additional **AdaCQR** baseline for both sparse and dense retrievers.

---

> ### Author Response · Authors · 2025-11-19
>
> `Q2. What is the latency analysis, for both data generation and inference? The method relies heavily on LLMs.`
>
> We appreciate the reviewer’s attention to latency and computational cost. DualReform introduces additional computation **only in the offline training pipeline**, **in exchange for removing the need for costly manual annotation of reference passages**. Once the CQR model is trained, **inference-time** latency is effectively **identical** to that of the underlying CQR baseline, since DualReform does not invoke any extra LLM calls or additional retrieval steps at test time.
>
> Compared to reference-based methods such as RetPO [Yoon et al.,2024], the extra cost in the training stage comes from three components: (i) response refinement via CQR (Eq. (9)), (ii) a retrieval step to obtain pseudo reference passages (Eq. (8)), and (iii) DPO training on the resulting pseudo reference passages (Eq. (10)). As summarized in Table R5, the overall overhead required by these steps is **moderate** and largely dictated by the available GPU resources. In particular, using a **single** NVIDIA RTX A6000 necessitates only an additional **6.41 hours**, with the main overhead stemming from stages (i) and (iii).
>
>
> |    Table R3       | (i) Response Refinement via CQR | | (ii) Retrieval of Pseudo Reference Passages | | (iii) DPO Training | |
> |------------------|--------------|-------------|-------------|--------------|----------|--|
> |      | A6000  | A100 (est.) | Sparse Retrieval  | Dense Retrieval | A6000  | A100 (est.)    |
> | Wall-clock time | 3.58h (1.289s/query)|1.77h (0.639s/query)| 1.15mins (0.015s/query)| 2.42mins (0.007s/query)  | 2.79h | 1.43h|
>
> *Table R5*: Wall-clock time for each major component of DualReform on SciConvQA. Time is reported in hours (h), minutes (min), and seconds (s) for a single RTX A6000 GPU. The "A100 (est.)" columns denote approximate wall-clock times on an NVIDIA A100 GPU, obtained by scaling the A6000 measurements using a 2.02× speedup factor following prior work [Massed Compute, 2024].
>
> ---
> [Mo et al.,2024a] CHIQ: Contextual History Enhancement for Improving Query Rewriting in Conversational Search. ACL, 2024
>
> [Mo et al.,2024b] History-Aware Conversational Dense Retrieval. ACL, 2024
>
> [Mo et al.,2023] Learning to Relate to Previous Turns in Conversational Search. KDD, 2023
>
> [Yoon et al.,2024] Ask Optimal Questions: Aligning Large Language Models with Retriever’s Preference in Conversational Search. arXiv, 2024
>
> [Zhao et al., 2024] WildChat: 1M ChatGPT Interaction Logs in the Wild. EMNLP, 2024
>
> [Ding et al., 2024] Enhancing Chat Language Models by Scaling High-quality Instructional Conversations. arXiv, 2023
>
> [Zheng et al., 2024] LMSYS-Chat-1M: A Large-Scale Real-World LLM Conversation Dataset. arXiv, 2023
>
> [Lai et al., 2025] AdaCQR: Enhancing Query Reformulation for Conversational Search via Sparse and Dense Retrieval Alignment. Coling, 2025
>
> [Massed Compute, 2024] LLama 3 Benchmark Across Various GPU Types. Massed Compute. 2024. Available at: https://massedcompute.com/llama-3-benchmark-across-various-gpu-types/
>
> [Ni et al., 2022] Large Dual Encoders Are Generalizable Retrievers. EMNLP, 2022
>
> [Xiong et al., 2020] Approximate nearest neighbor negative contrastive learning for dense text retrieval. arXiv, 2020
>
> [Anantha et al., 2021] Open-Domain Question Answering Goes Conversational via Question Rewriting. NACCL, 2021

---

> > ### Author Response · Authors · 2025-11-25
> > **Kindly Follow-Up**
> >
> > Dear Reviewer UcSN:
> >
> > We greatly appreciate your thoughtful feedback and the time devoted to this review, which substantially improved the quality of our work. If possible, we would be grateful for any additional insights at your convenience, as they would help us further improve this work. The authors would also be happy to discuss any aspects in more detail. Thank you very much.
> >
> > Sincerely,
> >
> > Authors

---

> > > ### Author Response · Authors · 2025-12-01
> > > **For Area Chair: Summary of Responses to Reviewer UcSN**
> > >
> > > Please note that many of the comments are **factually incorrect** and appear to have been **copied almost verbatim** from the review of our previous submission. We believe this reflects a lack of proper attention to the current manuscript, especially considering that the other three reviewers correctly understood the main motivation and contributions of our work. We respectfully request that the final decision not be unduly influenced by this unsubstantiated and unprofessional review.
> > >
> > > | Comment | Concern | Response |
> > > |---|---|---|
> > > | W1, Q3 | Insufficient motivation | **Unfactual.** Contrary to the reviewer's opinion, most modern conversational datasets do **not** have reference passages. See **Table R1**. More importantly, our problem setting aligns with the growing prevalence of LLM-based chat systems, where large-scale conversationals are collected without reference passages. |
> > > | W2 | Unfair comparison | **Misunderstanding.** The baseline setting is **totally fair** because the baselines require reference passages. |
> > > | W2 | Unfair comparison | **Misunderstanding.** The comparison is **totally fair** because the **same** dense retriever is used for **all** methods; we additionally report the results with an alternative retriever. See **Table R2**. |
> > > | W3 | Insufficient evaluation | **Misunderstanding.** The mentioned evaluation does exist in the original manuscript (Table 3). |
> > > | W3 | Missing baseline | **Additional results.** We provide additional results with the **most recent** baseline, AdaCQR. See **Table R4**. |
> > > | Q1 | Unclear difference | **Misunderstanding.** Aligned with W1. |
> > > | Q2 | Cost/latency analysis | **Additional results.** The cost of DualReform is very **small**. See **Table R5**. |
> > >
> > >
> > > | Dataset              | **#Conversations (K)** | **Reference passages** | Characteristics                                              |
> > > |----------------------|------------------------|------------------------|------------------------------------------------------------------|
> > > | **Recent large-scale conversational datasets** ||||
> > > | WildChat-4.8M-Full   | 4,800                  | ✗                      | In-the-wild ChatGPT logs across diverse domains                  |
> > > | UltraChat            | 1,400                  | ✗                      | Multi-turn dialogues mimicking user scenarios          |
> > > | LMSYS-Chat-1M        | 1,000                  | ✗                      | Chatbot Arena logs over multiple LLMs                       |
> > > | ShareGPT             | 90                     | ✗                      | User-shared ChatGPT conversations                                |
> > > | OASST1               | 66                     | ✗                      | Crowd-sourced dialogues with human ratings       |
> > > | *[All-Future-Datasets]*| ∞                      | ✗ *(rare)*               | *[Continuously growing with new domains and emerging topics]*  |
> > > | **Conversational search benchmarks** ||||
> > > | QReCC                | 13.7                   | ✓                      | Combined from QuAC, TREC CAsT, and NQ             |
> > > | TopiOCQA             | 3.5                    | ✓                      | Topics grounded in Wikipedia                          |
> > >
> > > *Table R1*. Recent large-scale conversational datasets versus conversational search benchmarks, showing the number of conversations and whether gold reference passages are annotated (K = thousands of conversations; ✓/✗ indicate the presence/absence of annotated reference passages). **In short, it is almost infeasible to annotate large-scale datasets.**

---

### Author Response · Authors · 2025-11-25
**For Area Chair: Summary of Discussion and Revisions**

Dear Area Chair,

We sincerely thank the Area Chair for the time and attention devoted to our submission. To facilitate your evaluation, we provide a brief summary of the discussion across all reviewers. For completeness, we have also summarized the reviewer-specific, detailed responses under each corresponding discussion thread.

First, we would like to express our gratitude for the reviewers’ ***positive assessment***, particularly their recognition of **DualReform’s strong empirical effectiveness** (All reviewers), achieving performance close to reference-based upper bounds through a **self-reinforcing, reference-free preference optimization framework** (Reviewers UcSN, YCPm, Kn6Q). We also appreciate the acknowledgement of the **comprehensive evaluation**, encompassing extensive comparisons, ablations, and theoretical analysis (Reviewers YCPm, Kn6Q, m8aF).

At the same time, we have endeavored to address the concerns raised by all reviewers. However, regarding **Reviewer UcSN**, many of his/her comments are **factually inaccurate**. Most notably, the reviewer dismisses *our "reference-free" motivation* simply because some datasets include reference passages. Yet, as clarified in *Table R1* and in the *discussion thread*, the **vast majority** of large-scale conversational datasets (outnumbering those with references by at least **a factor of 400**) do **not** provide reference passages, which directly motivates our approach. *We sincerely hope that the Area Chair would carefully evaluate the reviewer’s other claims together with our responses in the discussion thread, as they similarly stem from the incorrect premise.*


For all other reviewers’ comments, we have fully addressed them by **clarifying the problem setup and overall exposition** and **conducting several additional analyses**, as summarized in the table below. For each reviewer's comment, the corresponding author response has been posted as an *official comment* under the respective review on OpenReview. In parallel, the *manuscript* has been revised accordingly; the affected sections are listed in the “Revised Parts in Manuscript” column and highlighted in blue in the revised manuscript.



| Actions Taken for Reviewers' Comments                                                                                                     | Revised Parts in Manuscript |
|---------------------------------------------------------------------------------------------------------------|-----------------------------|
| ***Conceptual clarifications***                                                                                               |                             |
| Clarification of the notion of reference passages and practical scenarios in which they are unavailable      | Section 2.2, Appendix A.1                 |
| Clarification of the notion and scope of the reference-free setting                                        | Appendix A.2                 |
| ***Additional analyses***                                                                                       |                             |
| Evaluation with the ANCE dense retriever                                                                      | Section 5.1, Appendix F.7                |
| Evaluation with an additional CQR baseline                                                                    | Section 5.2                 |
| Evaluation of self-generated query reformulations in place of externally LLM-generated reformulations        | Section 5.4.2                 |
| Analysis of the effect of additional DualReform optimization iterations                                       | Appendix F.3                |
| Evaluation of generated responses using reference-free LLMEval metrics                                        | Section 5.1, 5.4.1                 |
| Component-level ablation of DualReform                                                                     | Appendix F.8                |
| Analysis of computational cost and latency                                                                    | Section 5.3.5                 |
| Empirical assessment of assumption validity                                                               | Appendix B.1                 |


Once again, we sincerely thank the Area Chair for the careful consideration given to our submission. We hope that the revisions and clarifications we have provided will be helpful in the evaluation process and support a favorable assessment of the paper.

Best regards,

Authors

---

### Meta-Review · Area_Chair_foaM · 2026-01-05

**Summary:**

The paper proposes DUALREFORM, a framework designed to address the lack of human-annotated reference passages in Conversational Query Reformulation (CQR). The authors introduce a "Dual-Role" mechanism where the CQR model acts not only as a query rewriter but also refines system responses to generate pseudo-labels. These labels are then used for iterative Direct Preference Optimization (DPO). By evaluating on datasets like QReCC, TopiOCQA, and a newly constructed synthetic dataset SciConvQA, the work demonstrates that this reference-free approach can outperform zero-shot baselines and approach fully supervised upper bounds.
**Rebuttal Conclusion**
Despite the authors' hard work and a comprehensive rebuttal that addressed specific technical gaps (e.g., adding strong baselines like AdaCQR), thegeneral agreement leans towards rejection due to the incremental nature of the methodological contribution and concerns regarding the problem framing. While the method is empirically effective, it is fundamentally an application of iterative self-training (bootstrapping) with a specific data augmentation technique. Consequently, the AC views this as a solid engineering improvement better suited for a specialized Information Retrieval conference rather than a fundamental advance in representation learning required for ICLR.

**Reviewer Concerns:**

**Concerns Addressed by the Rebuttal**:
**Missing Baselines and State-of-the-Art Comparison**: Resolved by the inclusion of the latest SOTA method (AdaCQR, 2025) and additional experiments with the ANCE retriever (Tables R11/R12). This addressed Reviewer m8aF's concern about how new the comparisons were.
**Dependence on External LLMs**: Resolved by the new "Self-Generated" experiment (Table R6), where the authors demonstrated that DUALREFORM can be trained using its own reformulations without relying on ChatGPT, addressing Reviewer YCPm's concern about cost and independence.
**Experimental Fairness (Partial Resolution)**: Addressed by the "Single-Role" ablation study (Table 3), which showed that the "Dual-Role" mechanism outperforms standard self-training. This technically counters Reviewer UcSN's claim that performance gains were only due to domain adaptation, though the underlying intuition about unfair comparison settings remains a valid criticism of the paper's framing.

**Outstanding Concerns**:
**Incremental Methodological Novelty**: This remains the primary reason for rejection. The core contribution—using a model to refine labels for its own training—is a well-established paradigm (Self-Training/Bootstrapping) in semi-supervised learning. Reviewer Kn6Q and the AC note that applying this to CQR with a specific heuristic represents an incremental engineering application rather than a methodological breakthrough.
**Marginal Utility vs. Complexity**: The "Dual-Role" mechanism introduces significant pipeline complexity (dual-task fine-tuning, prompt engineering) for what is ultimately a marginal gain over simpler, scalable self-training methods using modern LLMs. The paper does not convincingly argue that this complexity is justified by the performance delta.
**Problem Framing ("Reference-Free" vs. Weak Supervision)**: Reviewer Kn6Q noted ambiguity in the "Reference-Free" terminology. Methodologically, the work is classic Weakly Supervised Learning using pseudo-labels. Rebranding this as a new "Reference-Free" paradigm creates an issue of over-claiming the novelty of the setting.
**Reliance on Synthetic Benchmarks**: A substantial portion of the evaluation relies on SciConvQA, a dataset entirely created by GPT-4o. High performance here risks measuring the model's ability to fit errors or patterns of the generator rather than true generalization in noisy, real-world retrieval scenarios.

**Reviewer Scores:**

I think the scores are 2, 4, 6, 6.
Reviewer UcSN (2) maintained a strong rejection based on the experimental setup's fairness. Reviewer Kn6Q (4) remained negative due to concerns about theoretical clarity and definitions. Reviewers m8aF (6) and YCPm (6) were more positive regarding the empirical results. However, as AC, I must weigh the limited fundamental novelty heavily against the engineering metrics. The work is solid but incremental, fitting the profile of a "Reject" for a top-tier representation learning conference.

---

### Decision · Program_Chairs · 2026-01-26

Reject